# Cyst-independent oocyte phagocytosis builds the female reproductive reserve in mice

Yan Zhang[1]✉, Yingnan Bo[1], Kaixin Cheng[1], Ge Wang[1], Lu Mu[1], Jing Liang[1], Lingyu Li[1], Kaiying Geng[1], Xuebing Yang[1], Xindi Hu[1], Wenji Wang[2], Longzhong Jia[1], Xueqiang Xu[1], Jingmei Hu[3], Chao Wang[1], Fengchao Wang ⓘD [4], Yuwen Ke[1], Guoliang Xia[1] & Hua Zhang ⓘD[1]✉

## Abstract

During ovariogenesis, more than two-thirds of germ cells are sacrificed to improve the quality of the remaining oocytes. However, the detailed mechanisms behind this selection process are not fully understood in mammals. Here, we developed a high-resolution, four-dimensional ovariogenesis imaging system to track the progression of oocyte fate determination in live mouse ovaries. Through this, we identified a cyst-independent oocyte phagocytosis mechanism that plays a key role in determining oocyte survival. We found that oocytes act as individual cells, rather than connected cyst structures, during ovarian reserve construction. In this process, dominant oocytes capture and absorb cell debris from sacrificed oocytes to enrich their cytoplasm and support their survival. Single-cell sequencing indicated that the sacrificed oocytes are regulated by autophagy. When oocyte sacrifice was inhibited using autophagy inhibitors, the pool of surviving oocytes expanded, but they failed to fully develop and contribute to fertility. Our study suggests that mammals have evolved a cyst-independent selection system to improve oocyte quality, which is essential for sustaining a long reproductive lifespan.

Keywords Female Fertility; Ovarian Reserve; Oocyte Phagocytosis; Oocytes; Live Organ Imaging
Subject Categories Autophagy & Cell Death; Development

See also: K Wassmann

## Introduction

In sexually reproductive organisms, life begins with the combination of an oocyte and sperm, and successful fertilization requires high-quality gametes, especially the oocyte (Eppig, 2001). Therefore, females have developed a complex system for improving oocyte quality by sacrificing part of the oocytes to nurse the selected oocytes during ovariogenesis (De Cuevas et al, 1997; Lei and Spradling, 2016). This selection is believed to be a highly programmed and conserved process in both invertebrate and vertebrate species (Spradling et al, 2022). To understand this process, a large body of studies have focused on oogenesis in invertebrate model species, and the classic model of oocyte selection is derived from *Drosophila*. In this model, germ cells rapidly divide and form germline cyst structures in which multiple sister oocytes are cytoplasmically continuous owing to incomplete cytokinesis(De Cuevas et al, 1997; Pepling and C. Spradling, 1998). Organelle-enriched cytoplasm is then transported in an orderly manner from nursing germ cells to the dominant one through cytoplasmic bridges, which forms dominant oocyte that can contribute to fertility (Bolívar et al, 2001; Cox and Spradling, 2003; Spradling et al, 2022). In this model, oocyte connections and communications through intercellular bridges (IBs) in the cysts are believed to be indispensable for the determination of oocyte fate (De Cuevas et al, 1997; Spradling et al, 2022).

The life cycle of oocytes in mammals is much more complex than that in invertebrate species. During ovariogenesis in the fetal and neonatal periods, a small portion of the germ cells are selected while the other 2/3 of the total germ cells are sacrificed, resulting in the construction of a non-renewable ovarian reserve (Findlay et al, 2015; Lei and Spradling, 2013). After the selection, oocytes must survive for more than 1 year in rodents and ~50 years in humans to maintain the normal reproductive lifespan (Zhang et al, 2014). By tracing the oocyte development via a non-specific inducible cell labelling strategy in mice, Lei et al, showed that the organelle-enriched cytoplasm is transported from nursing germ cells to dominant cells through IBs in cysts, which ultimately determines the surviving oocytes to construct the ovarian reserve (Ikami et al, 2023; Lei and Spradling, 2013; Lei and Spradling, 2016). Recently, Niu et al, described a detailed model of cytoplasm exchanges in mouse ovarian cysts and investigated the inner molecular mechanisms involved in this process by single-cell RNA sequencing. These studies provide important information for understanding the model of early oocyte selection and development and demonstrate that the formation of oocyte cysts and the communications of sister oocytes in cysts are essential for their fate determination (Niu and Spradling, 2022). All of these studies

---

[1]State Key Laboratory of Animal Biotech Breeding, College of Biological Sciences, China Agricultural University, Beijing 100193, China. [2]School of Life Science, Taizhou University, Taizhou, Zhejiang 318000, China. [3]Center for Reproductive Medicine, Shandong University, Jinan, Shandong 250012, China. [4]National Institute of Biological Sciences, Beijing, China. ✉E-mail: yanzhang1011@cau.edu.cn; huazhang@cau.edu.cn

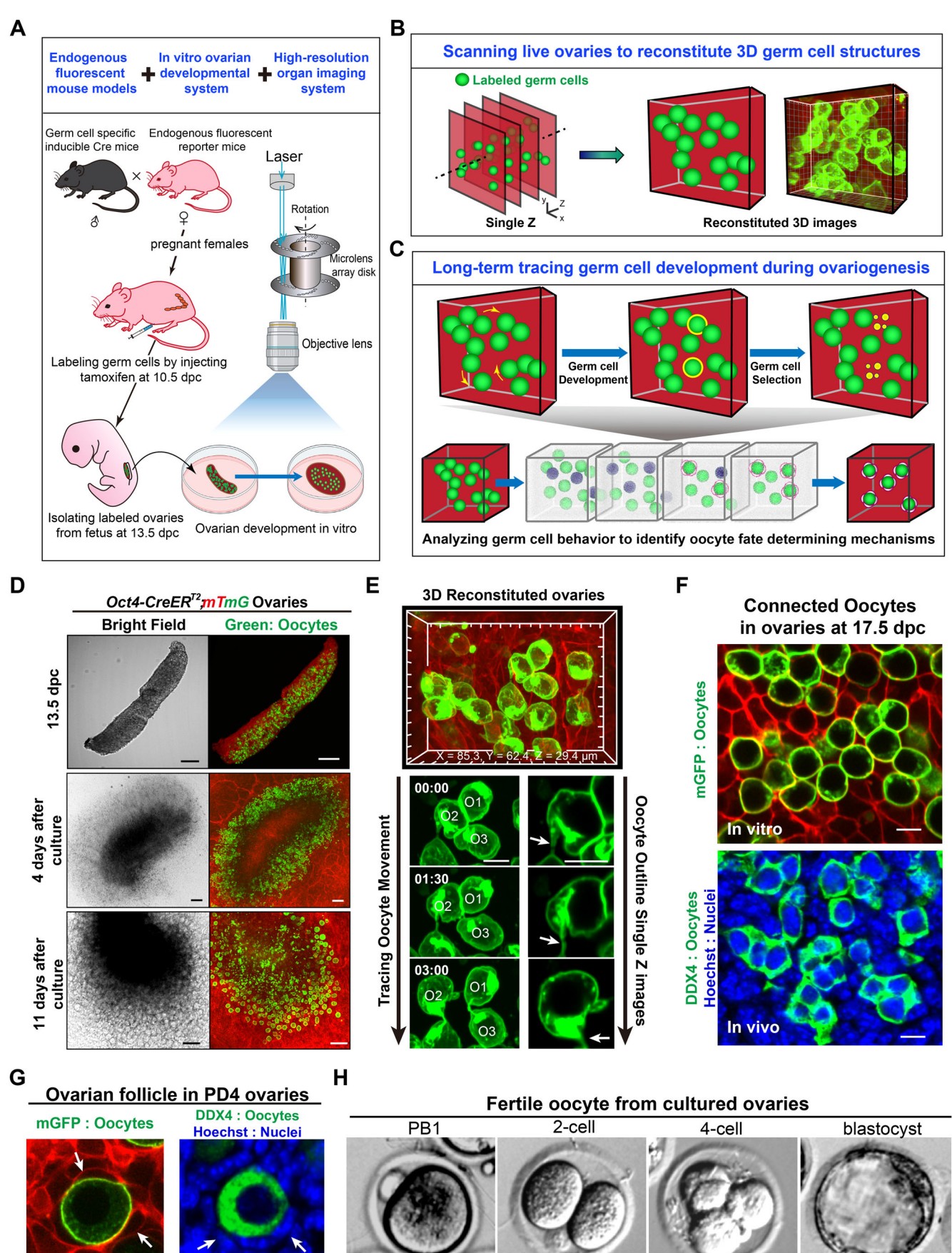

**A**

Endogenous fluorescent mouse models + In vitro ovarian developmental system + High-resolution organ imaging system

Germ cell specific inducible Cre mice
Endogenous fluorescent reporter mice
♂ × ♀
pregnant females
Labeling germ cells by injecting tamoxifen at 10.5 dpc
Isolating labeled ovaries from fetus at 13.5 dpc
Ovarian development in vitro

Laser
Rotation
Microlens array disk
Objective lens

**B** Scanning live ovaries to reconstitute 3D germ cell structures

● Labeled germ cells
Single Z
Reconstituted 3D images

**C** Long-term tracing germ cell development during ovariogenesis

Germ cell Development → Germ cell Selection

Analyzing germ cell behavior to identify oocyte fate determining mechanisms

**D** *Oct4-CreER^{T2};mTmG* Ovaries

Bright Field | Green: Oocytes

13.5 dpc
4 days after culture
11 days after culture

**E** 3D Reconstituted ovaries

X = 85.3, Y = 62.4, Z = 29.4 µm

Tracing Oocyte Movement
00:00 O2 O1 O3
01:30 O2 O1 O3
03:00 O2 O1 O3
Oocyte Outline Single Z images

**F** Connected Oocytes in ovaries at 17.5 dpc

mGFP : Oocytes
In vitro

DDX4 : Oocytes
Hoechst : Nuclei
In vivo

**G** Ovarian follicle in PD4 ovaries

mGFP : Oocytes
In vitro

DDX4 : Oocytes
Hoechst : Nuclei
In vivo

**H** Fertile oocyte from cultured ovaries

PB1 | 2-cell | 4-cell | blastocyst

**Figure 1. Establishing a 4D imaging system for recording germ cell development in live ovaries.**

(A) The schematic diagram illustrates the strategy for achieving high-resolution 4D imaging of germ cell development in fetal ovaries in vitro. The fetal ovaries, which contained labeled germ cells, were isolated and cultured in vitro, and then imaged under a high-resolution imaging system to record the developmental dynamics for long-term tracing. (B, C) Schematic diagrams illustrate the reconstitution of 4D imaging of germ cell development in live ovaries. The ovaries were scanned to form Z-stack images with detailed information, which were then merged to reconstitute high-resolution, 3D ovarian images containing detailed information on germ cell morphology and spatial location (B). After long-term culture with continuous imaging, the 3D images at different time points were merged to form 4D developmental dynamics (C). (D) The morphology of in vitro developed ovaries at different periods. The ovaries were isolated from $Oct4\text{-}CreER^{T2};mTmG$ females, in which the oocytes were labeled by mGFP (green) and other somatic cells were labeled by mTomato (red), at 13.5 dpc (top). After 4 days of culture, the ovaries were stabilized for imaging (middle), and a large number of developed oocytes survived in the ovaries after 11 days of culture (bottom). Scale bar: 100 μm. (E) The reconstituted 3D morphology of a group of oocytes (top). By tracing the developmental dynamics of these oocytes (bottom), the movement (left) and detail outline changes of these oocytes including subcellular structures (arrows) were recorded (right). Scale bar: 10 μm. (F) A single optical slide of cultured $Oct4\text{-}CreER^{T2};mTmG$ ovaries at the starting point of imaging, showing oocytes connected together forming cyst-like structures similar to those in fresh ovaries at 17.5 dpc. Scale bar: 10 μm. (G) At the ending point of imaging, the oocyte was covered by somatic granulosa cells (arrows) and formed a follicle similar to that in fresh ovaries at PD4. Scale bar: 10 μm. (H) In vitro fertilization of oocytes which were isolated from transplanted culture-ovaries. The oocytes were capable of forming healthy embryos at the two-cell, four-cell, and blastocyst stages. Scale bar: 20 μm.

highlighted that the early selection of oocytes follows a conserved model in mammals as in flies.

However, the fertile phenotype in the *Tex14* knockout mouse model, in which females lacked IBs, suggested that the cyst structure might not be indispensable for oocyte fate determination and the cyst-dependent germ cell model might not be the only mechanism for the oocyte selection in mice (Greenbaum et al, 2009; Ikami et al, 2021). Considering that the longevity of mammalian oocytes is much longer than that of invertebrate species, whether mammals have evolved any other specific strategies of oocyte selection to boost the quality of surviving oocytes for fitting their long reproductive lifespan remains elusive. Although there is a wealth of evidence from in vivo and in vitro studies indicating the role of cyst structures in regulating the fate of oocytes (Spradling et al, 2022), there has been a lack of long-term, detailed live observation and tracking of early oocyte development within the ovary. This has left us with limited knowledge of the specific mechanisms and related cell behaviors involved in mammalian oocyte selection.

In our study, we utilized endogenous germ cell reporter mouse models, as well as a live ovarian culture system and high-resolution imaging technology, to develop a subcellular resolution 4D whole ovary imaging system. Through this system, we were able to capture the entire process of oocyte selection in mouse ovaries and generate comprehensive visual data demonstrating a separated oocyte phagocytosis, which is unrelated to cysts, to select the best oocyte candidates for long-term survival. These findings provide a holistic understanding of how female mammals establish a high-quality oocyte reserve over time to accommodate their reproductive lifespan.

## Results

### Four-dimensional imaging to record germ cell development in the live mouse ovaries

To record the live behavior of germ cells during selection, we established a four-dimensional (4D) organ imaging platform by combining an in vitro mouse ovarian developmental system with a 3D high-resolution live organ imaging system (Fig. 1A–C). In the system, we scanned fetal ovaries containing endogenous fluorescence-labeled germ cells under a low phototoxicity high-

depth imaging system (Fig. 1A) and reconstituted 3D high-resolution images that contained detailed information about the morphology and spatial locations of germ cells in live ovaries (Fig. 1B). The ovaries were then cultured in vitro and regularly imaged to create 4D germ cell developmental dynamics covering the entire process of germ cell selection and follicle formation (Fig. 1C). This finally generated time-lapse images containing detailed information about behavior and developmental dynamics of germ cells during their fate determination in live ovaries (Movie EV1).

Since the major process of germ cell selection occurs between 17.5 dpc (day post coitum) to PD4 (postnatal day) in mice (Pepling, 2006), we designed a strategy to collect ovaries at 13.5 dpc (Fig. 1D, top) and culture them for 4 days to stabilize the tissue (Fig. 1D, middle). Because most germ cells have entered into meiosis after 17.5 dpc in the mouse ovaries (Pepling, 2006), the germ cells were termed oocytes in this study. The ovaries were then imaged for 7 consecutive days using the platform until follicle formation (Fig. 1D, bottom). To ensure the resolution of the oocytes, we initially imaged the ovaries, which were isolated from an oocyte endogenous multi-fluorescent mouse model (Greder et al, 2012; Zhang et al, 2014), $Oct4\text{-}CreER^{T2};mTmG$ mice, in which the membranes of all the oocytes were specifically labeled by membrane-GFP (mG) (Appendix Fig. S1) after tamoxifen treatment at 10.5 dpc. The continuous expression of endogenous, membrane-localized GFP ensures subcellular resolution for identifying any cellular structures or behaviors in live oocytes (Zhang et al, 2021) or ovaries (Xu et al, 2022). This allowed us to precisely identify the movement of oocytes (Fig. 1E, bottom left) and any changes in the outline of oocytes, including changes in the subcellular membrane structure (Fig. 1E, bottom right, arrows) in the 3D reconstituted ovaries (Fig. 1E).

Validating experiments revealed that the system well-supported ovarian organogenesis and follicle formation in vitro (Fig. EV1A,B). At the starting point of imaging (13.5 dpc + 4 days, equivalent to 17.5 dpc, and defined as c-17.5 dpc), we observed oocyte cyst structures (Fig. 1F, top) in the imaged ovaries, which were similar in morphology to oocytes in fresh ovaries at 17.5 dpc (Fig. 1F, bottom). At the endpoint of imaging (13.5 dpc + 11 days, equivalent to PD4 and defined as c-PD4), we observed ovarian follicles consisting of oocytes and surrounding (pre)granulosa cells (Fig. 1G, arrows), indicating the establishment of ovarian reserves in the cultured ovaries. Further allo-transplantation experiments

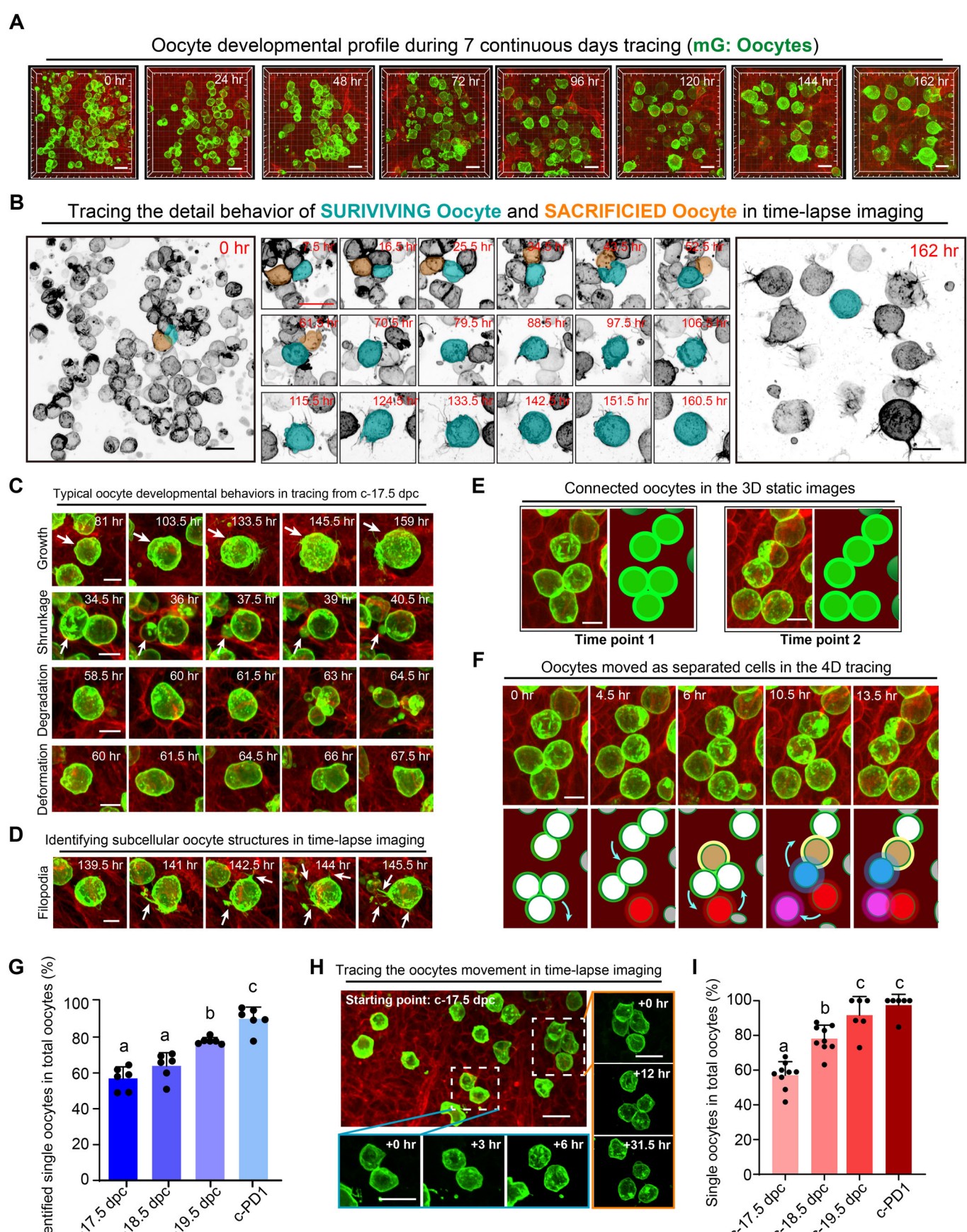

**A** Oocyte developmental profile during 7 continuous days tracing (mG: Oocytes)

0 hr  24 hr  48 hr  72 hr  96 hr  120 hr  144 hr  162 hr

**B** Tracing the detail behavior of SURIVIVING Oocyte and SACRIFICIED Oocyte in time-lapse imaging

0 hr

7.5 hr  16.5 hr  25.5 hr  34.5 hr  43.5 hr  52.5 hr
61.5 hr  70.5 hr  79.5 hr  88.5 hr  97.5 hr  106.5 hr
115.5 hr  124.5 hr  133.5 hr  142.5 hr  151.5 hr  160.5 hr

162 hr

**C** Typical oocyte developmental behaviors in tracing from c-17.5 dpc

Growth: 81 hr  103.5 hr  133.5 hr  145.5 hr  159 hr

Shrinkage: 34.5 hr  36 hr  37.5 hr  39 hr  40.5 hr

Degradation: 58.5 hr  60 hr  61.5 hr  63 hr  64.5 hr

Deformation: 60 hr  61.5 hr  64.5 hr  66 hr  67.5 hr

**D** Identifying subcellular oocyte structures in time-lapse imaging

Filopodia: 139.5 hr  141 hr  142.5 hr  144 hr  145.5 hr

**E** Connected oocytes in the 3D static images

Time point 1  Time point 2

**F** Oocytes moved as separated cells in the 4D tracing

0 hr  4.5 hr  6 hr  10.5 hr  13.5 hr

**G**

Identified single oocytes in total oocytes (%)

a  a  b  c

c-17.5 dpc  c-18.5 dpc  c-19.5 dpc  c-PD1

**H** Tracing the oocytes movement in time-lapse imaging

Starting point: c-17.5 dpc

+0 hr  +12 hr  +31.5 hr

+0 hr  +3 hr  +6 hr

**I**

Single oocytes in total oocytes (%)

a  b  c  c

c-17.5 dpc  c-18.5 dpc  c-19.5 dpc  c-PD1

Figure 2. Oocytes act as separated single cells during their final fate determination.

(A) A representative time-lapse image showing the oocyte developmental dynamics in live ovaries in vitro over a period of 162 h, from c-17.5 dpc to c-PD4. Scale bar: 20 μm. (B) Tracing the developmental dynamics of oocytes in the time-lapse image. All oocytes were inverted to black/white (b/w), and a survival oocyte was colored cyan, while a sacrificed oocyte was colored orange to highlight their developmental dynamics. Scale bar: 20 μm. All images of the surviving oocyte development are shown in Appendix Fig. S2. (C) Representative images showing the typical behaviors of oocytes during development, including growth (arrows in lane 1), shrinkage (arrows in lane 2), degradation and deformation. Scale bar: 10 μm. (D) Representative images showing the filopodia formation (arrows) of oocytes during development. Green, oocytes; Red, somatic cells. Scale bar: 10 μm. (E) 3D static images showing that oocytes are connected to cyst-like structures at any detected time points. Green, oocyte; Red, somatic cells. Scale bar: 10 μm. (F) Tracing the oocyte movement in the 4D tracing images (from the time point 1 to point 2 of Fig. 2E). Showing connected oocytes moved in different directions, and most of the cyst-like structures were actually assembled by separated single oocytes in the ovaries after c-17.5 dpc. Cyan arrows: oocyte movement direction. Scale bar: 10 μm. (G) Quantifying the ratio of separated oocytes in total oocytes from c-17.5 dpc to c-PD1. By continuously tracing the oocyte movement in the 4D time-lapse imaging, the ratio of oocytes with separation events in total oocytes was counted every 24 h. Data were presented as the mean ± SD. $n = 6$ ovaries ($n > 40$ oocytes per ovary). c-18.5 dpc vs. c-17.5 dpc: $p$ value = 0.213; c-19.5 dpc vs. c-17.5 dpc: $p$ value = 0.0003; c-PD1 vs. c-17.5 dpc: $p$ value = 5.20E-06; c-19.5 dpc vs. c-18.5 dpc: $p$ value = 0.0033; c-PD1 vs. c-18.5 dpc: $p$ value = 7.60E-05; c-PD1 vs. c-19.5 dpc: $p$ value = 0.011. (H) Identifying the movement of labeled oocytes in the Oct4-CreER$^{T2}$;mTmG ovaries after low dosage of tamoxifen treatment. Only a small portion of cysts were labeled in the ovaries, with most of them breaking down into single oocytes at c-17.5 dpc. Green, oocytes; Red, somatic cells. Scale bar: 20 μm. (I) The ratio of single oocytes in total labeled oocytes from c-17.5 dpc to c-PD1 in ovaries with low labeling density of oocytes. Data were presented as the mean ± SD. $n \geq 6$ ovaries ($n > 15$ oocytes per ovary). c-18.5 dpc vs. c-17.5 dpc: $p$ value = 3.01E-05; c-19.5 dpc vs. c-17.5 dpc: $p$ value = 0.0001; c-PD1 vs. c-17.5 dpc: $p$ value = 7.64E-08; c-19.5 dpc vs. c-18.5 dpc: $p$ value = 0.0208; c-PD1 vs. c-18.5 dpc: $p$ value = 0.0007; c-PD1 vs. c-19.5 dpc: $p$ value = 0.608. Statistical significance was determined by unpaired one-way ANOVA tests. $P$ (a, b) < 0.05, $P$ (a, c) < 0.05, $P$ (b, c) < 0.05. Source data are available online for this figure.

showed that the follicles in the in vitro-developed ovaries were healthy for growth (Fig. EV1A) and able to ovulate fertilized oocytes (Fig. 1H), demonstrating that our system supported a normal process of oocyte development and ovarian reserve construction.

## Oocytes act as single cells for their final fate determination

After validation, we traced oocyte development in Oct4-CreER$^{T2}$;mTmG ovaries to analyze their detailed behaviors during selection. In total, 1099 EGFP oocytes in six labeled ovaries were imaged from c-17.5 dpc and traced for 162 h with 1.5 h interval (Fig. 2A; Appendix Fig. S2; Movie EV1). Time-lapse tracing successfully recorded the complete developmental dynamics of 594 oocytes, and statistical analysis revealed that the majority of the traced oocytes were eliminated (Fig. 2B, orange), whereas 14.7 ± 2.7% of the oocytes survived until c-PD4 (Fig. 2B, cyan) to form ovarian follicles. By analyzing the 4D dynamics of oocyte development, we recorded distinct developmental fates of different oocytes, including growth to survive (Fig. 2C, Lane 1, arrows), shrinking to elimination (Fig. 2C, Lane 2, arrows) and degradation into small pieces (Fig. 2C, Lane 3). Furthermore, subcellular changes of oocytes, such as oocyte deformation (Fig. 2C, Lane 4) and formation of oocyte filopodia (Fig. 2D, arrows), were also detectable in the images. Statistical analysis suggested that these events or behaviors represented distinguished dynamics with temporal characteristics during the selection of oocytes (Appendix Fig. S3A–D). Thus, time-lapse images provide a comprehensive and visual database for analyzing oocyte fate determination in live ovaries.

According to the classic model, oocyte fate determination is dependent on oocyte communication within the cyst (Lei and Spradling, 2016). To investigate this, we first traced the development of oocytes within one cyst to identify any behavioral differences. Despite a high density of labeled oocytes crowding the ovaries (Fig. 2B), cyst-like structures with different numbers of connected oocytes (Fig. 2E) were still identifiable in 3D static images at all detected time points. However, unexpectedly, analysis of the 4D tracing revealed that the oocytes within the identified 3D cysts were not stably connected. As the ovaries developed, the

oocytes were actively moving, and the attached oocytes in one cyst separated and moved in different directions to connect with other oocytes, forming a new group of cysts (Fig. 2F; Movie EV2). By tracing the oocyte movement in 4D imaging, we found that the separation and re-connection events were frequently observed in cyst-like structures with the development of the ovaries. Statistical analysis revealed that the proportions of oocytes exhibiting separation behaviors gradually increased (Fig. 2G) with ovarian development, with over 80% of oocytes moving as single cells at c-19.5. This finding suggested that most of the cyst structures observed in the static images were temporary oocyte attachments from 17.5 dpc onward.

To confirm our findings, we reduced the tamoxifen dose and labeled a small portion of germ cells (18.5 ± 2.3%, Fig. EV2A) at 11.5 dpc to trace germ cell development in Oct4-CreER$^{T2}$;mTmG mice. This strategy allowed us to trace the developmental trajectory of germ cells from distinguishable cysts (Fig. EV2B) with no disruption from surrounding cysts. Using the 4D imaging platform, we traced the movement of these labeled germ cells from c-17.5 dpc to c-PD1 (Fig. EV2C), and the tracing results clearly showed that most of the germ cells (57.1 ± 7.8%) were moving as single cells in the ovaries since c-17.5 dpc, and most of the connected oocytes also separated in the following 48 h (Fig. 2H). Statistical analysis revealed that the percentage of single oocytes increased from 57.1 ± 7.8% at c-17.5 dpc to 78.1 ± 7.7% at c-18.5 dpc and 91.6 ± 10.8% at c-19.5 dpc (Figs. 2I and EV2D). By c-PD1, almost all the oocytes (97.5 ± 6.2%) had separated into single cells, and few cysts with connected oocytes existed in the ovaries (Fig. 2I). This finding demonstrated that cyst breakdown was completed much earlier than follicle formation in the mouse ovary and oocytes acted as independent cells rather than as cysts in the last period of their fate determination. Hence, an uncovered cyst-independent system is necessary to complete the oocyte fate determination in the mouse ovaries.

## Cell phagocytosis between individual oocytes determines the selection of surviving oocytes

To identify the model of cyst-independent system for oocyte fate determination, we focused on the oocytes that formed ovarian

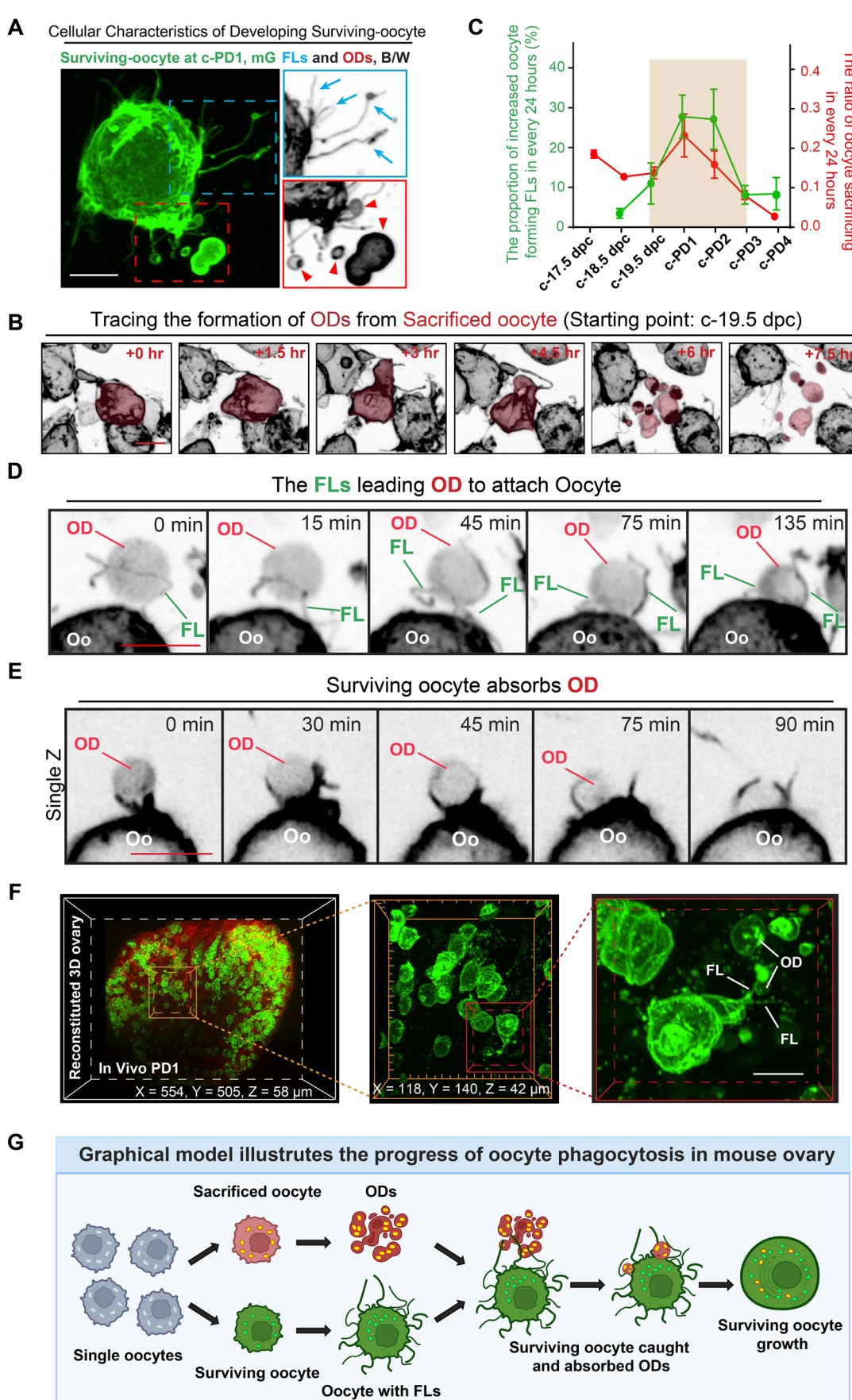

**A** Cellular Characteristics of Developing Surviving-oocyte

Surviving-oocyte at c-PD1, mG **FLs** and **ODs**, B/W

**C**

**B** Tracing the formation of ODs from Sacrificed oocyte (Starting point: c-19.5 dpc)

+0 hr  +1.5 hr  +3 hr  +4.5 hr  +6 hr  +7.5 hr

**D** The **FLs** leading **OD** to attach Oocyte

0 min  15 min  45 min  75 min  135 min

**E** Surviving oocyte absorbs **OD**

Single Z

0 min  30 min  45 min  75 min  90 min

**F**

Reconstituted 3D ovary

In Vivo PD1   X = 554, Y = 505, Z = 58 μm   X = 118, Y = 140, Z = 42 μm

**G** Graphical model illustrutes the progress of oocyte phagocytosis in mouse ovary

Sacrificed oocyte   ODs

Single oocytes   Surviving oocyte   Oocyte with FLs   Surviving oocyte caught and absorbed ODs   Surviving oocyte growth

**Figure 3. Intensive separated oocytes phagocytosis occurs to decide their fate.**

(A) A representative image showing the developing cellular characteristics of surviving oocytes. Highlighting the formation of filopodia-like structures (FLs, arrows) and surrounded by various oocyte debris (OD, arrowheads). Scale bar: 10 µm. (B) Tracing the derivation of ODs. The sacrificed oocyte was colored red to highlight the progress of OD formation. Scale bar: 10 µm. (C) The frequency of surviving oocytes forming FLs (green line) and sacrificed oocytes forming ODs (red line), showing a consistent peaking period of two events (brown region) from c-19.5 dpc to c-PD2. Data were presented as the mean ± SEM. n = 4 ovaries (more than 40 oocytes in each time point per ovary). (D) The FLs from surviving oocytes extended towards the ODs to assist them in moving towards the surviving oocyte and then attached ODs. Scale bar: 10 µm. (E) Representative time-lapse images showing the detailed progress of how surviving oocytes absorbed the ODs. Scale bar: 10 µm. (F) FLs on the oocytes and ODs were observed in fresh transparent ovaries at PD1. Scale bar: 10 µm. (G) The graphic model illustrates the cyst-independent oocyte phagocytosis in mouse ovaries. The sacrificed oocyte sacrifices itself to form various ODs containing organelles, enriched cytoplasm. The ODs are caught by the surviving oocyte formed FLs to move forward to the surviving oocyte. Then the surviving oocyte absorbs the ODs to enrich the cytoplasm and organelles for survival. Green, surviving oocyte; Red, sacrificed oocyte; Dark red, ODs. Source data are available online for this figure.

follicles at the end of culture and tracked backwards to identify their developmental characteristics of these oocytes at earlier stages (Appendix Fig. S4A). Through time-lapse images, we discovered several key developmental characteristics of the cyst-independent oocyte selection process. First, all the surviving oocytes (a total of 84 oocytes) formed filopodia-like structures (FLs) on their cell surface during their development (Fig. 3A, arrows, blue box). Second, multiple oocyte debris (ODs) with GFP membrane were observed surrounding the surviving oocytes, particularly during the period of FL formation (Fig. 3A, arrowheads, red box). By tracing the origin of ODs, we observed that some oocytes broke into small vesicles with a complete membrane (Fig. 3B, Oocyte with red color, Movie EV3). The counting results demonstrated that the formation of ODs peaked from c-19.5 to c-PD2 (Fig. 3C, red line), which corresponded to the peak period of surviving oocytes forming FLs (Fig. 3C, green line). These observations suggest an interesting model of how single oocytes determine their fate, wherein the fate determination of oocyte elimination and survival through short-range interaction between separated oocytes, similar to cell phagocytosis (Hoijman et al, 2021), occurs between separated oocytes. We therefore termed this process as oocyte phagocytosis, in which surviving oocytes engulf sacrificed oocyte-derived ODs through phagocytosis to enrich their cytoplasm.

To confirm our proposed model and capture the detailed oocyte behaviors during oocyte phagocytosis, we increased the frequency of photography to every 15 min for analyzing the oocyte development from c-PD1 over a 12-h period (Movie EV4). The resulting 4D tracing images revealed that surviving oocytes extended the FLs towards the surrounding ODs in a dynamic manner (Appendix Fig. S4B,C), directing the ODs toward themselves (Fig. 3D, Appendix Fig. S4D; Movies EV4–6). Subsequently, surviving oocytes absorbed the ODs, facilitated by process structures in their membrane (Fig. 3E; Appendix Fig. S4D,E; Movie EV4), resulting in a noticeable increase in oocyte size due to cytoplasm enrichment (Appendix Fig. S4F). Moreover, by re-analyzing the long-term 4D imaging to calculate the engulfment events of each surviving oocyte during their development (Appendix Fig. S5A), we found that a strong correlation between the frequency of engulfment events per oocyte and the fate of oocyte survival. Surviving oocytes that formed follicles had to engulf at least 5 ODs, with an average of 15 engulfment events during their development. In sharp contrast, no oocytes that engulfed fewer than 4 ODs survived (Appendix Fig. S5B). These findings provide evidence that surviving oocytes 'eat' ODs through phagocytosis, and engulfing a sufficient number of ODs is essential for the survival of selected oocytes.

Our findings provide evidence of oocyte phagocytosis within cultured mouse ovaries. However, it remains uncertain whether this phenomenon is a naturally occurring physiological process. To confirm this point, we utilized a whole-mount ovarian transparent imaging system (Xu et al, 2022) and transmission electron microscopy (TEM) imaging to examine the presence of FLs and ODs in fresh Oct4-CreER$^{T2}$;mTmG ovaries at PD1 (Figs. 3F, left and EV3). Through 3D reconstituted ovarian images and TEM images, we discovered that oocytes possess irregular cell shapes, much like those found in cultured ovaries. Importantly, we identified both FLs on oocytes and ODs in the ovaries (Fig. 3F, middle and right) at high resolution, as well as oocyte-extended FLs oriented toward ODs (Fig. 3F, right). Additionally, TEM imaging revealed phagocytosis between the oocyte and OD (Fig. EV3). Furthermore, we observed the engulfment of ODs by surviving oocytes in freshly isolated Oct4-CreER$^{T2}$;mTmG ovaries at PD1 (Movie EV7). These experimental evidences clearly confirm that oocyte phagocytosis is a physiological event that occurs within mouse ovaries. Therefore, our results demonstrate that intense oocyte phagocytosis occurs between different single oocytes, and the surviving oocytes appropriate the enriched organelles from ODs to enhance their quality for survival (Fig. 3G).

## Oocyte phagocytosis leads to cytoplasm exchanges, nurse the surviving oocytes

Our findings demonstrated that surviving oocytes engulf ODs through phagocytosis to enrich their cytoplasm, but it is unknown whether this is the primary strategy for cytoplasm exchange beyond cyst communication. To investigate this, we examined the strategy of oocyte cytoplasm exchange by introducing an oocyte cytoplasm tracing mouse model, Oct4-CreER$^{T2}$;Rainbow mice, into the system. In the ovaries of Oct4-CreER$^{T2}$;Rainbow females (Appendix Fig. S6A, B), the germ cells randomly expressed CFP or RFP in their cytoplasm (Fig. 4A), therefore cytoplasm exchange could be directly visualized through mixed fluorescence in cells (Fig. 4A, arrow). By monitoring the color of oocytes, we observed a small portion of oocytes with mixed fluorescent cytoplasm (purple) at all time points from c-17.5 to c-PD4 (Fig. 4B, arrows), and the ratio of purple oocytes showed that the peak of cytoplasm exchange occurred from c-PD1 to c-PD3 (Fig. 4C), following the peak of FL and OD formation. This finding suggested that oocyte phagocytosis is the primary, if not the only, mechanism of cytoplasm exchange between individual oocytes.

In the classic model, oocyte connections are essential for cytoplasm exchanges (Spradling et al, 2022). Given the high

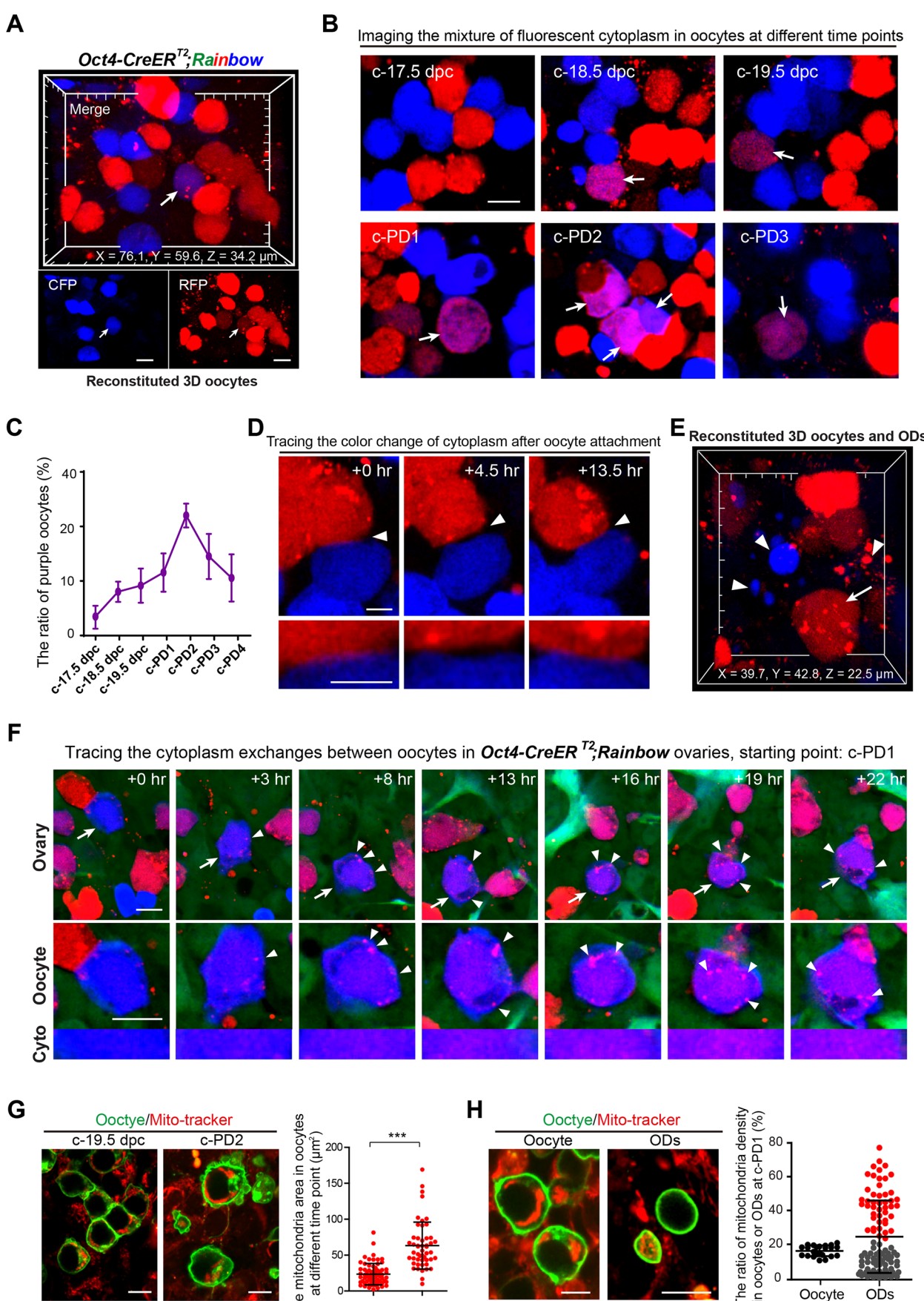

**A** Oct4-CreER^T2;Rainbow

Merge

X = 76.1, Y = 59.6, Z = 34.2 μm

CFP          RFP

Reconstituted 3D oocytes

**B** Imaging the mixture of fluorescent cytoplasm in oocytes at different time points

c-17.5 dpc     c-18.5 dpc     c-19.5 dpc

c-PD1          c-PD2          c-PD3

**C**

The ratio of purple oocytes (%)

c-17.5 dpc, c-18.5 dpc, c-19.5 dpc, c-PD1, c-PD2, c-PD3, c-PD4

**D** Tracing the color change of cytoplasm after oocyte attachment

+0 hr     +4.5 hr     +13.5 hr

**E** Reconstituted 3D oocytes and ODs

X = 39.7, Y = 42.8, Z = 22.5 μm

**F** Tracing the cytoplasm exchanges between oocytes in *Oct4-CreER^T2;Rainbow* ovaries, starting point: c-PD1

+0 hr    +3 hr    +8 hr    +13 hr    +16 hr    +19 hr    +22 hr

Ovary

Oocyte

Cyto

**G** Ooctye/Mito-tracker

c-19.5 dpc     c-PD2

The mitochondria area in oocytes at different time point (μm²)

***

c-19.5 dpc     c-PD2

**H** Ooctye/Mito-tracker

Oocyte          ODs

The ratio of mitochondria density in oocytes or ODs at c-PD1 (%)

Oocyte     ODs

Figure 4. Oocyte phagocytosis is the major strategy for cytoplasm exchanges after cyst breakdown.

(A) 3D reconstructed images of *Oct4-CreER^T2^;Rainbow* ovaries are shown, displaying random expressions of RFP (red) or CFP (blue) in oocyte cytoplasm. Cytoplasm exchange leads to a mixture of fluorescence in the oocyte (purple, arrow). Scale bar: 10 μm. (B) Analysis of oocyte fluorescence reveals an increased number of oocytes with mixed fluorescent cytoplasm (purple, arrows) from c-17.5 dpc to c-PD3. Scale bar: 10 μm. (C) The quantification of the ratio of purple oocytes in total oocytes from c-17.5 dpc to c-PD4, showing that the peak of cytoplasm exchanges occurs between c-PD1 to c-PD3. Data were presented as the mean ± SD. *n* = 4 ovaries. More than 200 oocytes were measured at each time point. (D) Tracing the attachment between oocytes with different fluorescence in the time-lapse imaging. No cytoplasm exchange occurs after attachments (arrowheads) between oocytes. Scale bar: 5 μm. (E) The reconstructed 3D images demonstrate the existence of ODs (arrowheads) surrounding the oocyte (arrow) in the *Oct4-CreER^T2^;Rainbow* ovaries. (F) The time-lapse images trace the progress of a blue surviving oocyte (arrow) absorbing red ODs (arrowheads), leading to a gradual change of cytoplasm color from blue to purple. The bottom line highlights the color change of the surviving oocyte cytoplasm. Scale bar: 10 μm. (G) Analyzing the absolute area of mitochondria in oocytes (red: mitochondria, green: oocytes). Showing a significantly increased average mitochondria area in oocytes at c-PD2 compared to that in oocytes at c-19.5 dpc. Scale bar: 10 μm. More than 50 oocytes were measured at each time point. Data were presented as the mean ± SD. *p* value = 3.21E-11. ***$P < 0.001$, by two-tailed unpaired Student's *t*-test. (H) Comparing the relative density of mitochondria in oocytes and ODs (red: mitochondria, green: oocytes or ODs) at c-PD1 (left). A great variation of mitochondria density was observed in ODs but not in oocytes (right). Scale bar: 10 μm. Oocytes: *n* = 23; ODs: *n* = 107. Data were presented as the mean ± SD. Source data are available online for this figure.

frequency of oocyte attachments and reassembly observed after cyst breakdown, we examined whether cytoplasm exchange occurred through connections with attached, separated oocytes. By analyzing high magnification images, we traced a total of 100 attachments between oocytes with different colors (Fig. 4D, arrowheads), but did not observe any mixing of cytoplasm after the connections of oocytes (Movie EV8). Furthermore, we observed small colorful ODs (Fig. 4E, arrowheads) that surrounded some oocytes (Fig. 4E, arrow), and 4D tracing analysis clearly captured that oocytes with a single color fluorescent protein (Fig. 4F, blue oocyte, arrow) absorbed various ODs with another color fluorescence (Fig. 4F, red ODs, arrowheads and Movie EV9) which led to the progressive color change of the oocyte from blue to purple (Fig. 4F, Cyto). Meanwhile, by suppressing the formation of FL using Formin inhibitor SMIFH2 (Nishimura et al, 2021), we found that a dramatic decrease in the ratio of purple oocytes formed at c-PD2. This finding confirms that cytoplasm exchanges rely on oocyte phagocytosis (Appendix Fig. S7A–D). Together, our data clearly demonstrate that cytoplasm exchange does not rely on connections and attachments between oocytes after the cyst breakdown, suggesting that oocyte phagocytosis is the only strategy for cytoplasm and organelle enrichment during the final period of oocyte selection.

Corroborating this observation, we identified mitochondria in the oocytes in *Oct4-CreER^T2^;mTmG* ovaries, and we noted that the oocytes presented a marked increase in mitochondrial numbers and density from c-19.5 dpc to c-PD2 (Fig. 4G; Appendix Fig. S8), which is the period of peak OD and FL formation. Furthermore, quantitative analysis revealed considerable variation in mitochondrial density among different ODs (Fig. 4H), suggesting that not only the cytoplasm but also organelles, such as mitochondria, were enriched in sacrificed oocyte-derived ODs and transferred to surviving oocytes.

## Single-cell sequencing suggests that phagocytosis is regulated by autophagy

To dissect the underlying mechanisms that control oocyte phagocytosis, we performed single-cell RNA-seq to identify the molecular expressing characteristics of oocytes at PD1, the time at which intensive phagocytosis occurs in ovaries (Fig. 5A). A total of 1307 oocytes expressing an average of 4084 genes in PD1 ovaries were retained for analysis after stringent quality control. The highly

variable genes were subsequently selected for perform a principal component analysis (PCA) and KNN clustering, and the oocytes were separated into six clusters (C0 to C5) on the basis of their gene expressing profiles (Fig. 5B). We found that meiotic related genes including *Dmc1* (Bishop, 1994), *Meiob* (Luo et al, 2013) and *Spo11* (Romanienko and Camerini-Otero, 2000) are highly expressed in the C0 and C1, suggesting that these oocytes are in active meiotic progress and should be in an early developmental state (Fig. 5C,D, red; Dataset EV1). We therefore defined them as the candidate oocytes that are waiting to be selected. Clusters 2 and 3 are highly expressed *Sohlh1* (Pangas et al, 2006), *Ooep* (Pierre et al, 2007) and *Dppa3* (Bortvin et al, 2004), which are the specific genes of oocytes in formed follicles (Fig. 5C, D, green and Dataset EV1), indicating that these cells were selected to be the surviving oocytes for forming follicles in ovaries. Further analyses identified that *Diaph3* (Formin) (Schirenbeck et al, 2005), *Epn2* (Epsin) (Aguilar et al, 2006) *Vil1* (Villin) (Friederich et al, 1989; Huelsmann et al, 2013) and *Rdx* (Zhang et al, 2021), which are genes involved in the regulation of microvilli or filopodia formation (Fig. 5C,D blue; Appendix Fig. S9A; Dataset EV1), are highly expressions in C1, C2, and C3. These findings are consistent with the cell behaviors of surviving oocytes in our 4D imaging. Additionally, by counting the proportion of surviving oocytes in the imaging data at c-PD1, we found that the proportion of surviving oocytes was identical between the sequencing (C2 + C3: 35.1%) and the imaging data (35.2 ± 6.13%). In Clusters 4 and 5, oocyte death-related genes, including *Trp53* (Cochrane et al, 2020), *Anxa5* (Ghislat et al, 2012) and *Vdac1* (Vijayan et al, 2022) were highly expressed (Fig. 5C,D, purple; Dataset EV1), implying that they were sacrificed oocytes.

Consistent with gene expression profiles, lineage trajectory reconstruction according to the highly variable genes predicted that a clear transcriptional map of oocyte fate determination (Fig. 5E), i.e. the candidate oocyte population (C0 and C1, red state) differentiated into two distinguished cell lineages which were the surviving oocytes (C1, C2, and C3, green state) and sacrificed oocytes (C4 and C5, purple state). To further analyze how lineages acquire their identities, we selected branch-related genes and analyzed the dynamics of gene expression along the predicted pseudotime from the candidate lineage to the surviving oocytes or sacrificed oocytes (Appendix Fig. S9B). As shown in Appendix Fig. S9B,C, 399 genes were highly expressed in the candidate population, but their expression levels were dramatically reduced after differentiation to the surviving or sacrificed populations

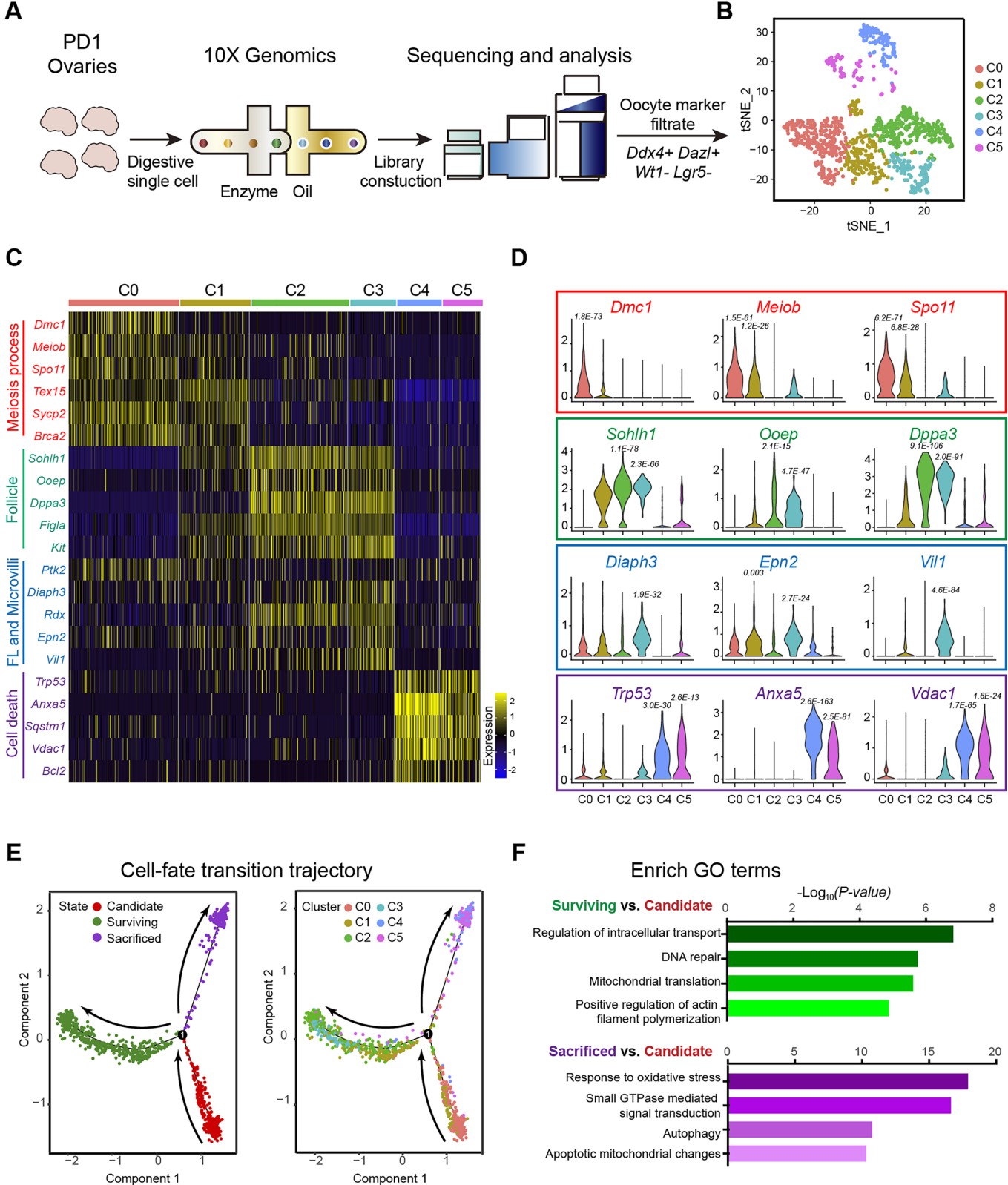

© The Author(s)

◄ **Figure 5.   Single-cell RNA-Seq to analyze molecular mechanisms in regulating oocyte phagocytosis.**

(A) The flowchart provided an overview of the scRNA-seq for analyzing the gene expressing profile of oocytes in PD1 ovaries. (B) T-distributed stochastic neighborhood embedding (t-SNE) plots divided oocytes in PD1 ovaries into six separate clusters. (C, D) The gene expression signatures of various oocyte clusters were shown through a heatmap (C) and violin plots (D). Meiosis process-related genes (red) gradually decrease from C0 to C5, with expression levels highest in the C0 and C1 clusters. Genes related to oocytes forming ovarian follicles (green) highly express in many oocytes in C2 to C3 clusters. FL and microvilli formation-related genes (blue) highly express in most oocytes across C0 to C3 clusters, with expression levels gradually rising from C0 to C3. Cell death-regulated genes (purple) highly express in most oocytes in C4 and C5. *P* value in (D) based on the Wilcoxon rank sum test. C0: $n = 354$; C1: $n = 225$; C2: $n = 313$; C3: $n = 146$; C4: $n = 142$; C5: $n = 127$. (E) Pseudotime prediction of the oocyte lineage trajectory, showing the cell lineage states (left), and cell clusters across the branches (right). The pseudotime scores were labeled by arrows. Cell lineage analysis and pseudotime dissection predicted the candidate oocyte lineage (red state in the left panel, C0 and C1 in the right panel) transitioned to the surviving oocyte lineage (green state in the left panel, C1, C2, and C3 in the right panel) and sacrificed oocyte lineage (purple state in the left panel, C4 and C5 in the right panel). (F) Representative gene ontology terms enriched in the surviving oocyte lineage (green histograms) and sacrificed oocyte lineage (purple histograms) are highlighted. *P* value used by the hypergeometric test.

(Appendix Fig. S9B,C, red box; Dataset EV2). After differentiation, the expression levels of 696 genes, including *Nobox* (Rajkovic et al, 2004), *Uchl1* (Goto et al, 2015) and *Ybx2* (Zhang et al, 2023) (Appendix Fig. S9B,C, green box; Dataset EV2), increased toward the surviving oocyte lineage. The GO terms of these genes were enriched with "regulation of intracellular transport", "DNA repair", "mitochondrial translation" and "positive regulation of actin filament polymerization" (Fig. 5F, green bar-charts; Dataset EV3). Meanwhile, the expression of 1578 genes gradually increased with the differentiation of sacrificed oocyte lineage (Appendix Fig. S9B,C, purple box; Dataset EV4). These genes were enriched with GO terms on "response to oxidative stress", "small GTPase mediated signal transduction", "autophagy", and "apoptotic mitochondrial changes" (Fig. 5F, purple bar-charts; Dataset EV4), suggesting that these genes or pathways might be crucial for the regulation of sacrificed oocyte fate determination.

## Suppressing oocyte phagocytosis leads to a large but dysfunctional ovarian reserve in the ovaries

To determine the physiological significance of oocyte phagocytosis in the ovaries, we chose to block autophagy, the genes of which were dramatically differentially expressed in sacrificed oocytes in our scRNA-seq analysis, to disrupt the progress of oocyte phagocytosis. In the 4D imaging system, we added a series of well-tested autophagy inhibitors, including 3-MA (3-methyladenine) (Wu et al, 2013), MRT68921 (Su et al, 2022), Bafilomycin A1 (Shacka et al, 2006), and Chloroquine (Mauthe et al, 2018) from c-17.5 dpc, and traced the oocyte development and fate until c-PD4. In general, we found that the oocytes in all inhibitor-treated groups remained in a relatively silenced state with no active oocyte phagocytosis at c-PD1 (Figs. 6A and EV4A,B), and more oocytes survived at c-PD4 (Figs. 6A and EV4C,D). Using 3MA as a representative treatment, we performed detail cellular analysis of the oocyte phagocytosis in the *Oct4-CreER^{T2};mTmG* model, and found that the frequencies of both the behaviors of oocyte-forming FLs (Fig. 6B, arrows, C) and the formation of ODs (Fig. 6B, arrowheads) were dramatically lower in the treated ovaries than that in the control ovaries, leading to the failure of both growth (Fig. 6D) and death (Fig. 6E) of oocytes in the ovaries. Consistent with the inactivity of oocyte phagocytosis, we found significantly increased number of oocytes at the end of imaging of c-PD4 in 3-MA treated ovaries (Figs. 6F and EV4D, control 3768 ± 709 vs. 3-MA 5996 ± 386), with dramatically decreases in both the average oocyte size (Appendix Fig. S10A) and the density of mitochondria

in surviving oocytes (Appendix Fig. S10B) in the treated ovaries. These results demonstrated that a lack of oocyte phagocytosis suppressed the selection of oocytes and might decline the oocyte quality of ovarian reserve.

Next, we extended the time of in vitro culture to monitor the developmental capability of oocytes that lacked phagocytosis. After 7 days of additional culture (equal to PD 11), the oocytes in the control ovaries continued growing, and their average diameter increased to 52.4 ± 4.8 μm (Fig. 6G; Appendix Fig. S10C, control), showing our system well supports the growth of oocytes in vitro. However, the lacking phagocytosis oocytes just increased to 37.4 ± 6.1 μm (Fig. 6G; Appendix Fig. S10C, 3-MA) at the end of culture. These data demonstrated that the oocytes that did not undergo phagocytosis-related cytoplasm exchange were unable to grow fully. Meanwhile, we performed allo-transplantation to examine the in vivo developmental capability of the oocytes with or without phagocytosis (Fig. 6H), and we found that a fast loss of oocytes occurred in 3-MA treated ovaries after 2 weeks of transplantation (Fig. 6H, 3-MA + 2 weeks), and no healthy oocytes survived in the treated ovaries after 4 weeks of surgery (Fig. 6H, 3-MA + 4 weeks). This is in sharp contrast to the healthy development and ovulation of oocytes (Fig. 6H, control) in the control group after transplantation. Therefore, we concluded that oocyte phagocytosis is essential to boost oocyte quality for the maintenance of normal fertility in female mice.

In summary, our study suggests a two-step selection model to boost oocyte quality in mammals (Fig. 6I). After cyst formation in the fetal ovaries, the germ cells are initially selected through the IBs in the cysts (Niu and Spradling, 2022; Spradling et al, 2022) (Fig. 6I, step 1). With the cyst breakdown, intensive oocyte phagocytosis occurs between separated independent oocytes to complete the secondary selection, and finally determines the number of surviving oocytes that will form ovarian reserve (Fig. 6I, step 2). This specific oocyte phagocytosis system might be crucial for mammals to fit their long reproductive lifespan.

## Discussion

Oocytes are a specialized cell population responsible for generating new life and ensuring the continuation of a species, and complex processes of oogenesis have been evolved to ensure that ovulated oocytes are of the highest quality (Eppig, 2001; Hsueh et al, 2015). One commonly understood mechanism involves the transfer of cytoplasm and organelles from sister nurse oocytes to the selected

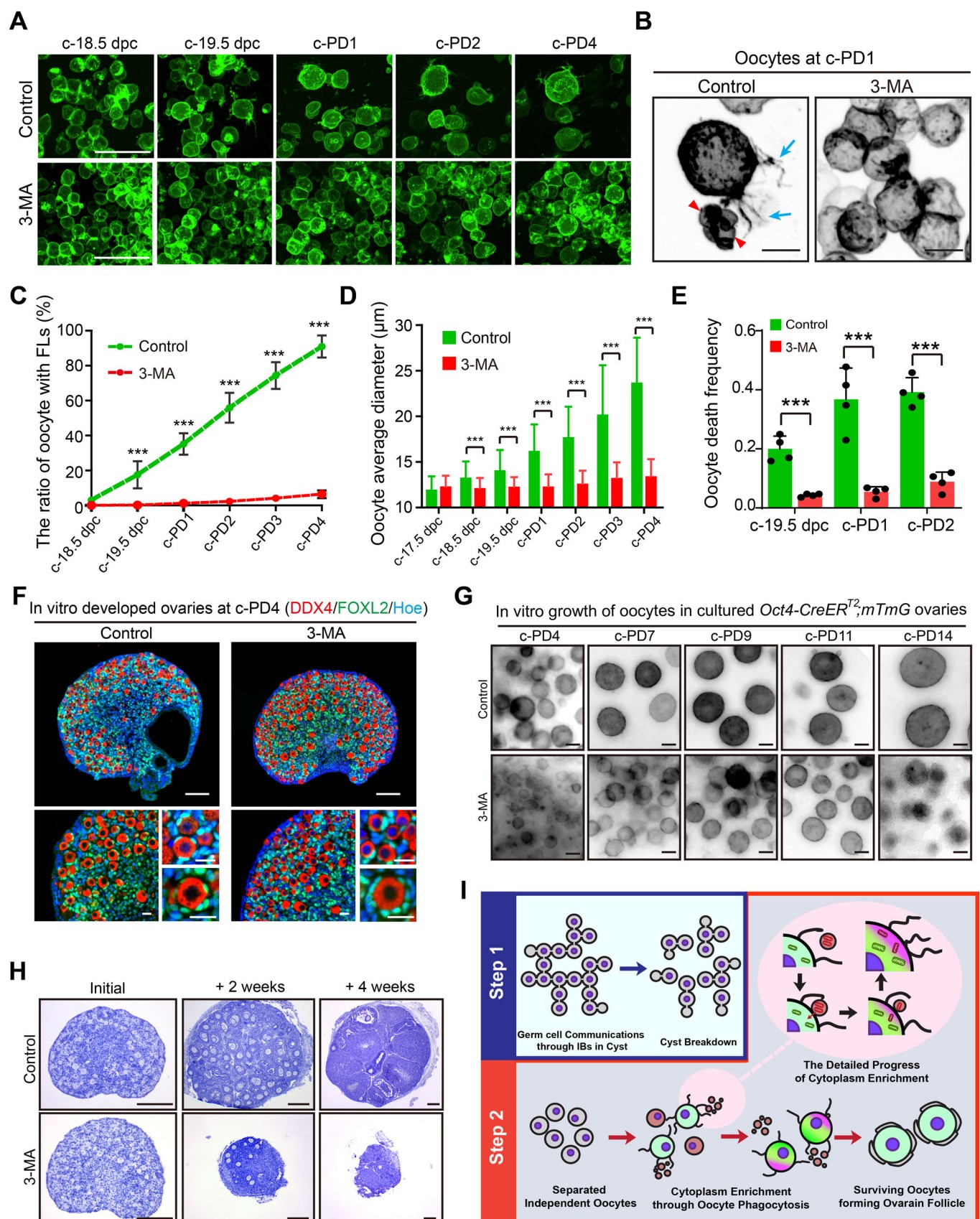

**A** c-18.5 dpc c-19.5 dpc c-PD1 c-PD2 c-PD4

Control / 3-MA

**B** Oocytes at c-PD1 — Control / 3-MA

**C** The ratio of oocyte with FLs (%) — Control, 3-MA
c-18.5 dpc, c-19.5 dpc, c-PD1, c-PD2, c-PD3, c-PD4

**D** Oocyte average diameter (μm) — Control, 3-MA
c-17.5 dpc, c-18.5 dpc, c-19.5 dpc, c-PD1, c-PD2, c-PD3, c-PD4

**E** Oocyte death frequency — Control, 3-MA
c-19.5 dpc, c-PD1, c-PD2

**F** In vitro developed ovaries at c-PD4 (DDX4/FOXL2/Hoe) — Control / 3-MA

**G** In vitro growth of oocytes in cultured *Oct4-CreER^T2;mTmG* ovaries
c-PD4, c-PD7, c-PD9, c-PD11, c-PD14 — Control / 3-MA

**H** Initial / + 2 weeks / + 4 weeks — Control / 3-MA

**I** Step 1 — Germ cell Communications through IBs in Cyst → Cyst Breakdown; The Detailed Progress of Cytoplasm Enrichment
Step 2 — Separated Independent Oocytes → Cytoplasm Enrichment through Oocyte Phagocytosis → Surviving Oocytes forming Ovarain Follicle

**Figure 6. Blocking oocyte phagocytosis leads to functional failure of ovarian reserve construction.**

(A) Tracing the oocyte development with or without 3-MA treatment. Normal oocyte phagocytosis and selection were observed in control ovaries, but not in the 3-MA-treated ovaries. More oocytes with a dramatically decreased size survived at the end of culture at c-PD4. Scale bar: 50 μm. (B) Representative images display the morphology of oocytes in 3-MA-treated ovaries and control ovaries at c-PD1. Surviving oocytes with FLs (arrows) and sacrificed oocyte-forming ODs (arrowheads) were observed in the control ovaries but not in 3-MA-treated ovaries. Scale bar: 10 μm. (C) Statistical analysis of the oocytes with FLs shows that only a few oocytes formed FLs in 3-MA-treated ovaries. More than 100 oocytes were measured at each time point. c-18.5 dpc: $p$ value = 0.0007; c-19.5 dpc: $p$ value = 0.001; c-PD1: $p$ value = 5.8E-06; c-PD2: $p$ value = 2.78E-06; c-PD3: $p$ value = 2.81E-07; c-PD4: $p$ value = 5.7E-09. (D) Quantification of oocyte average diameter shows significant growth retardation in oocytes from 3-MA-treated ovaries compared to those from control ovaries. More than 50 oocytes were measured at each time point. c-17.5 dpc: $p$ value = 0.5; c-18.5 dpc: $p$ value = 1.28E-06; c-19.5 dpc: $p$ value = 2.37E-11; c-PD1: $p$ value = 9.59E-24; c-PD2: $p$ value = 2.02E-27; c-PD3: $p$ value = 1.99E-18; c-PD4: $p$ value = 5.79E-20. (E) Statistical analysis demonstrates a dramatically decreased frequency of oocyte death in 3-MA-treated ovaries compared to that in control ovaries from c-19.5 dpc to c-PD2. $n$ = 4 ovaries in every group. c-19.5 dpc: $p$ value = 0.001; c-PD1: $p$ value = 4.01E-07; c-PD2: $p$ value = 6.35E-07. (F) Immunofluorescent staining displays ovarian morphology at c-PD4. More oocytes survived in 3-MA-treated ovaries compared to those in control ovaries. Magnified images show ovarian follicles formed in both the 3-MA group and the control group. Red: DDX4, Green: FOXL2, Blue: Hoechst. Scale bar (top): 100 μm; Scale bar (bottom): 20 μm. (G) Investigation of the growth capability of oocytes derived from ovaries with or without 3-MA treatment. Although 3-MA was removed, oocytes in 3-MA-treated ovaries were unable to fully grow. Scale bar: 10 μm. (H) Histological analysis of ovarian development after 3-MA treatment. Normal follicle development and corpus luteum formation were observed in control ovaries during 4 weeks of in vivo development. In the 3-MA group, only a few oocytes survived after 2 weeks of transplantation, and no healthy follicles were found in the ovaries at 4 weeks after surgery. Scale bar: 200 μm. (I) The model of two-step oocyte selection in mammals. In the first round of cyst-dependent selection, orderly cytoplasm exchange occurs in cysts between intercellular bridges to improve the quality of selected germ cells. After cyst breakdown, the second round of selection, which is cyst-independent, occurs through intense oocyte phagocytosis. The surviving oocytes form FLs to capture organelles enriched ODs, which are derived from sacrificed oocytes, to enhance their quality. The best oocytes survive through two-step oocyte selection, constructing the ovarian reserve that supports female fertility throughout their life. The colors were inverted to black/white (b/w) in (D) and (G) to highlight oocyte morphology. Data were presented as the mean ± SD. ***$P$ < 0.001, by two-tailed unpaired Student's $t$-test. Source data are available online for this figure.

oocytes, thereby enhancing their quality (Spradling et al, 2022). The classic model of cytoplasm exchange suggests that this process occurs in a gentle and orderly manner and is solely dependent on oocyte connections within the cyst. Since all germ cells in a cyst are derived from a single progenitor germ cell, it appears that germ cell selection operates like a "primogeniture system", where the original germ cell with the most intercellular bridges (IBs) is deemed the ultimate winner and develops into an ovulated egg (Lei and Spradling, 2016; Niu and Spradling, 2022; Soygur et al, 2021). This model has been well-identified in insects, especially in *Drosophila*, and recent studies suggest it may also apply to mice, indicating that this cyst-dependent communication model is conserved across invertebrates to vertebrate species, including mammals (Spradling et al, 2022).

Our current findings reveal a cyst-independent oocyte phago-cytosis system that ultimately determines the fate of oocyte survival in mice. These findings also suggest that a two-step germ cell selection model is responsible for constructing ovarian reserves in mammals. During ovariogenesis, germline cysts are formed following rapid division of primordial germ cells in fetal ovaries, leading to the first round of selection within the cyst (Niu and Spradling, 2022). In the perinatal ovaries, a second round of selection, which is previously unknown, is initiated by a highly intensive oocyte phagocytosis after the cyst breakdown into individual oocytes. By utilizing time-lapse tracing, we propose that the origin of this phagocytosis is derived from a burst of sacrificed oocytes, which is directly or indirectly regulated by autophagy. This leads to the redistribution of organelles into the cytoplasm of sacrificed oocytes and eventual formation of oocyte-derived debris (ODs). These ODs are then caught by surviving oocytes, which generate follicle-like structures (FLs), which assist the surviving oocytes merge ODs into cells to complete the enrichment of cytoplasm and organelles. After the oocyte phagocytosis, the quality and quantity of the ovarian reserve are established, which determines the reproductive resources available to support female fertility throughout their lifetimes.

Our study on oocyte phagocytosis revealed that cyst-dependent communication is neither the only nor the primary mechanism for cytoplasm enrichment during oocyte selection in mammals. Instead, specific oocyte phagocytosis during mammalian ovario-genesis appears to play an essential role in determining the quality of surviving oocytes. When oocyte phagocytosis is blocked by adding autophagy inhibitors, the surviving oocytes fail to enrich their cytoplasm with organelles, particularly mitochondria. While mitochondrial fission may also contribute to the increase in oocyte mitochondrial population during this stage of selection, it is clear that oocyte phagocytosis is one of the main mechanisms driving the enhancement of selected oocyte quality. This results in an enlarged yet incompetent oocyte pool, suggesting that oocyte phagocytosis is indispensable for enhancing oocyte quality in mammals. This conclusion is supported by previous observations (Greenbaum et al, 2009) and our current data that female mice with *Tex14* mutations, which disrupt the stability of IBs, exhibit a reduced ovarian reserve size but remain fertile. Despite the absence of the first round of selection that relies on IB communications, the second round of oocyte phagocytosis, which is cyst-independent, still occurs (Appendix Fig. S11). This ensures the construction of a functional ovarian reserve in females. These findings suggest that oocyte phagocytosis, which is a specific mechanism for oocyte selection in mice, plays a pivotal role in determining oocyte quality and female fertility.

Over the past decade, accumulating evidence has indicated that cell competition, whereby higher fitness cells outcompete and eliminate lower fitness cells, plays an important role in various biological events during mammalian embryonic development (Baker, 2020; Esteban-Martínez and Torres, 2021; Kim and Jain, 2020). This selection mechanism is characterized by a range of intense cellular behaviors, including the engulfment of loser cells by winner cells to enhance their own development (Li and Baker, 2007; Zhu et al, 2019). This is reminiscent of the phagocytic behaviors observed in this study. Interestingly, the key regulators of somatic cell competition, such as autophagy signaling

(Nagata et al, 2019) and *Tp53* (Bowling et al, 2018; Cochrane et al, 2020; Ghafari et al, 2009), have also been found to be involved in mammalian oocyte selection. This observation offers an intriguing perspective, suggesting that cyst-independent oocyte phagocytosis represents an atypical form of cell competition. In this process, surviving oocytes act as winner cells, engulfing oocyte debris (ODs) from "loser" oocytes, ultimately contributing to the establishment of the ovarian reserve in mammals. The unique nature of oocyte competition presents an exciting opportunity to explore the molecular networks that govern this process. A more comprehensive understanding of these mechanisms could offer valuable insights into the factors influencing oocyte quality and the maintenance of reproductive function in female mammals.

Regarding the mechanism of oocyte phagocytosis, or oocyte competition, our data suggest that autophagy plays a crucial role in regulating oocyte fate during this process, particularly in the sacrifice of oocytes. However, our findings also reveal the complexity of autophagy's involvement in oocyte phagocytosis. Notably, the upregulation of five ATG8 family proteins, including ATG3, is not restricted to core autophagy pathways but also extends to alternative mechanisms such as LC3-associated phagocytosis and responses to lysosomal damage (Jacomin et al, 2020). This suggests that these alternative autophagy-related pathways, rather than canonical autophagy, may be more directly involved in regulating the sacrificed oocytes. Furthermore, the use of broad-spectrum autophagy inhibitors complicates our interpretation, as they affect both surviving and sacrificed oocytes, thereby masking the specific role of autophagy in shaping the loser phenotype. To address these limitations, future studies employing autophagy-specific reporters (Yim et al, 2022) and genetic loss-of-function approaches targeting core autophagy factors (e.g., FIP200, ATG9a, and ATG14) (Zhao and Zhang, 2019) in sacrificed oocytes will be critical for definitively elucidating the role of autophagy in oocyte selection.

Beyond our findings on oocyte selection, our study also exposes fundamental limitations in traditional static imaging methods commonly used in the study of development in mammals. Unlike transparent model organisms such as *Drosophila* (King et al, 1968) or *C. elegans* (Corsi et al, 2015), mammalian tissues are opaque, restricting the ability to perform long-term live imaging. Consequently, mammalian developmental events have historically been inferred from static observations such as histological sectioning and staining, leading to fragmented temporal snapshots. Our continuous live imaging approach directly challenges these conventional assumptions, revealing that what has traditionally been considered cyst structure actually consists of individual oocytes during the core phase of oocyte fate determination. Furthermore, we demonstrated that oocyte selection in this stage does not depend on direct cytoplasmic exchange between cysts, as previously hypothesized. These findings underscore the potential biases introduced by inferring dynamic developmental mechanisms from static data. Moving forward, the development and application of advanced in vivo imaging technologies will be crucial for refining our understanding of mammalian development across various tissues.

Overall, our research has provided a dynamic depiction of the early development and fate determination of mammalian oocytes, revealing a previously unknown mechanism of preferential survival through intense oocyte phagocytosis during the selection process. The identification of this phagocytosis among oocytes in mice

indicates that the construction of the mammalian ovarian reserve is not merely adherent to the conventional cyst-dependent model, but rather suggests an alternative mechanism that strategically selects oocytes conducive to a prolonged reproductive lifespan.

# Methods

**Reagents and tools table**

| Reagent/resource | Reference or source | Identifier or catalog number |
| --- | --- | --- |
| **Experimental models** | | |
| C57BL/6N | The Laboratory Animal Center of the Institute of Genetics (Beijing) | N/A |
| ICR | The Laboratory Animal Center of the Institute of Genetics (Beijing) | N/A |
| *Rosa26*rbw/+ | Dr. Kui Liu's lab (Zhang et al, 2012) | N/A |
| *mTmG* | Jackson Laboratory | 007576 |
| *Oct4-CreER*T2 | Dr. Ping Zheng's lab (Greder et al, 2012) | N/A |
| *Tex14*-/- | Dr. Fengchao Wang's lab | N/A |
| **Antibodies** | | |
| Goat anti-FOXL2 Polyclonal | Novus Biologicals | IMG-3228 |
| Mouse anti-DDX4 monoclonal | Abcam | ab27591 |
| Rabbit anti-TOM20 | Cell Signaling Technology | #42406 |
| Donkey anti-Rabbit IgG (H + L) Highly Cross-Adsorbed Secondary Antibody, Alexa Fluor 488 | Invitrogen | A21206 |
| Donkey anti-Mouse IgG (H + L) Highly Cross-Adsorbed Secondary Antibody, Alexa Fluor 488 | Invitrogen | A21202 |
| Donkey anti-Goat IgG (H + L) Highly Cross-Adsorbed Secondary Antibody, Alexa Fluor 488 | Invitrogen | A11055 |
| Donkey anti-Rabbit IgG (H + L) Highly Cross-Adsorbed Secondary Antibody, Alexa Fluor 568 | Invitrogen | A10042 |
| Donkey anti-Mouse IgG (H + L) Highly Cross-Adsorbed Secondary Antibody, Alexa Fluor 568 | Invitrogen | A10037 |
| **Oligonucleotides and other sequence-based reagents** | | |
| PCR primers | This study | Method-Animals |
| **Chemicals, Enzymes and other reagents** | | |
| DMEM/F12 | GIBCO | 1320033 |
| Insulin-transferrin-sodium selenite medium | Invitrogen | I3146 |
| Penicillin-streptomycin | Invitrogen | 15140122 |

| Reagent/resource | Reference or source | Identifier or catalog number |
|---|---|---|
| Fetal bovine serum | GIBCO | 10437028 |
| Corn oil | Sigma-Aldrich | C8267 |
| Tamoxifen | Sigma-Aldrich | 75648 |
| Praformaldehyde | Santa Cruz | 30525-89-4 |
| hematoxylin | Santa Cruz | sc-24973A |
| donkey serum | Jackson ImmunoResearch | 017-000-121 |
| Hoechst 33342 | Sigma-Aldrich | B2261 |
| Pregnant mare serum gonadotropin | Sansheng Biological Technology | N/A |
| Human chorionic gonadotropin | Sansheng Biological Technology | N/A |
| Hyaluronidase | Sigma-Aldrich | H-3757 |
| HTF medium | Merk-Millipore | MR-70-D |
| KSOM medium | Merk-Millipore | MR-020P-5F |
| N-methylacetamide | Sigma-Aldrich | M26305 |
| Histodenz | Sigma-Aldrich | D2158 |
| 1-thioglycerol | Sigma-Aldrich | M1753 |
| MitoTracker™ Orange CMTMRos | Thermo Fisher Scientific | M7510 |
| Trypsin | Thermo Fisher Scientific | 25200056 |
| RNase-free DNase I | Thermo Fisher Scientific | EN0523 |
| Bovine serum albumin | Sigma-Aldrich | A7030 |
| 3-MA | Sigma-Aldrich | M9281 |
| MRT68921 | Selleckchem | S7949 |
| Bafilomycin A1 | MedChemExpress | HY-100558 |
| Chloroquine | MedChemExpress | HY-17589A |
| SMIFH2 | MedChemExpress | HY-16931 |
| **Software** | | |
| Imaris | OXFORD instruments | https://imaris.oxinst.com |
| ImageJ | N/A | https://imagej.net/ij/ |
| Fusion 2.1 | OXFORD instruments | https://andor.oxinst.com/products/dragonfly#fusion |
| Cell Ranger (v5.0.1) | N/A | https://github.com/10XGenomics/cellranger |
| R (version 4.1.0) | N/A | https://www.R-project.org/ |
| Seurat (10X analysis) | (Satija et al, 2015) | https://github.com/satijalab/seurat |
| Monocle (v2.18.0) | (Qiu et al, 2017) | http://cole-trapnell-lab.github.io/monocle-release/ |
| **Other** | | |

| Reagent/resource | Reference or source | Identifier or catalog number |
|---|---|---|
| NovaSeq 6000 | Illumina | N/A |
| AO/PI Cell Viability Kit | Logos biosystems | F23001 |
| SPRIselect Reagent Kit | Beckman Coulter | B23318 |
| Chromium Next GEM Single Cell 3′ Kit v3.1 | 10x Genomics | PN-1000268 |
| NEBNext® UltraTM RNA Library Prep Kit | NEB | E7530 |

## Animals

The C57 and ICR mice were obtained from the Laboratory Animal Center of the Institute of Genetics (Beijing).

The $Rosa26^{rbw/+}$ and $mTmG$ mice (007576, Jackson Laboratory) were generated as previously reported (Zhang et al, 2012; Zheng et al, 2014), and $Rosa26^{rbw/+}$ mice as a gift from Dr. Kui Liu. The $Oct4\text{-}CreER^{T2}$ mice (Greder et al, 2012) were gifted from Dr. Ping Zheng at Yunnan Key Laboratory of Animal Reproduction, Kunming Institute of Zoology, Chinese Academy of Sciences, Kunming, China.

In addition, to obtain $Oct4\text{-}CreER^{T2}$ mice, we designed polymerase chain reaction (PCR) primers for the Cre enzyme DNA sequence (forward 1: 5′- CCAAGGCAAGGGAGGTAG ACAAG -3′, forward 2: 5′- GCTTTCTCCAACCGCAGGCTC TC -3′, reverse: 5′- GC CCTCACATTGCCAAAAGACGG-3′). $Oct4\text{-}CreER^{T2}$ mice were crossed with $mTmG$ and $Rainbow$ mice to generate $Oct4\text{-}CreER^{T2};mTmG$ and $Oct4\text{-}CreER^{T2};Rainbow$.

The $Tex14$ knockout mice were generated in the Lab of Dr. Fengchao Wang at the National Institute of Biological Sciences, Beijing, China. The generation process involved designing guide RNAs (gRNAs) using the CRISPR guide-design tool (https://crispr.dbcls.jp/). The selected gRNA sequences are tgtgagtggacg-gacctccctgg and ccttcattacagcgctgacactc. Mouse zygotes, obtained by mating of males with superovulated C57BL/6J females, were injected with a mixture of Cas9 mRNA (80 ng/μl) and sgRNA (40 ng/μl) for gene modification. These injected zygotes were subsequently transferred into pseudopregnant ICR females to produce genetically modified offspring. To investigate the role of $Tex14$ and intercellular bridges (IBs) in oocyte phagocytosis, $Tex14$ knockout mice ($Tex14^{-/-}$) were crossed with $Oct4\text{-}CreER^{T2};mTmG$ mice. This crossbreeding resulted in $Tex14^{+/-};Oct4\text{-}CreER^{T2};mTmG$ mice after several rounds of breeding. To obtained the fetal $Tex14^{-/-};Oct4\text{-}CreER^{T2};mTmG$ ovaries, the $Tex14^{+/-};Oct4\text{-}CreER^{T2};mTmG$ males were crossed with either $Tex14^{-/-};Oct4\text{-}CreER^{T2};mTmG$ or $Tex14^{+/-};Oct4\text{-}CreER^{T2};mTmG$ females. After checking the plug, pregnant females were administered tamoxifen (TAM) at a dosage of 50 mg/kg body weight by intraperitoneal injection at 10.5 days post coitum (dpc). This treatment aimed to label oocyte membranes in fetal $Tex14^{-/-};Oct4\text{-}CreER^{T2};mTmG$ ovaries. In addition, to obtain $Tex14^{-/-}$ mice, we designed polymerase chain reaction (PCR) primers for genotyping (forward 1: 5′- AGA-GAATCTTCCTCCTTTATCTGTTTGGG -3′, reverse 1: 5′- GGGA AATCTCTTTCTGTGCTGATGGAC-3′, reverse 2: 5′- TCATTTG ATTCTCGCTGCTTTCCG-3′).

All mice were housed in mouse facilities under 16/8-h light/dark cycles at 26 °C and humidity 40–70% with access to chow and water ad libitum. The animal experiments conformed to the guidelines and regulatory standards of the Institutional Animal Care and Use Committee of China Agricultural University, No. AW72012202-3-1.

## Tamoxifen (Tam) administration to label oocytes

To collect the *Oct4-CreER^{T2};mTmG* and *Oct4-CreER^{T2};Rainbow* fetal ovaries, *mTmG* or *Rainbow* homogenous females at 6–8 weeks were mated with adult *Oct4-CreER^{T2}* males overnight. The presence of vaginal plugs in the following morning is counted as 0.5 dpc, and the day after partum is defined as PD1. Tamoxifen (Tam, 75648, Sigma-Aldrich) was resuspended in 95% (v/v) ethanol (100 mg/ml) and then diluted in corn oil (C8267, Sigma-Aldrich) to a final concentration of 20 mg/ml. To label the oocytes in the ovaries of *Oct4-CreER^{T2};mTmG* and *Oct4-CreER^{T2};Rainbow* females, Tam, at a high dosage of 50 mg·kg$^{-1}$ BW or low dosage of 20 mg·kg$^{-1}$ BW, was intraperitoneally injected into pregnant mice at 10.5 dpc. To assess the labeling efficiency in the ovaries, Tam treated *Oct4-CreER^{T2};mTmG* ovaries were sectioned and subjected to counterstaining using the DDX4 antibody. The efficiency of oocyte labeling was determined by calculating the ratio of GFP-positive oocytes to the total number of DDX4-positive cells (Fig. EV2A).

## In vitro development of fetal ovaries

The fetal ovaries were obtained from *Oct4-CreER^{T2};mTmG* and *Oct4-CreER^{T2};Rainbow* mouse fetuses at 13.5 dpc, and the mesonephros were separated and removed in precooled PBS (10 mM, pH 7.4) under a stereomicroscope (Stemi 305, Zeiss) with sterile conditions. To observe the detailed behaviors and development of oocytes, the fetal ovaries were cultured on 35 mm glass bottom dish (D35-14-1-N, Cellvis) with the DMEM/F12 (Dulbecco's Modified Eagle Medium/Nutrient Mixture F12, 1320033, Gibco) supplemented with 1% ITS (insulin-transferrin-sodium selenite medium, I3146, Invitrogen), 10% FBS (fetal bovine serum, GIBCO, Life Technologies) and penicillin-streptomycin (100 IU/ml; 15140122, Invitrogen). After 4 days of adherent culture, the fetal ovaries (c-17.5 dpc) were attached firmly to the bottom of dishes for live-cell imaging. The culture media was half-changed every other day to maintain the development of the ovaries. The culture-ovaries were photographed in a living cell workstation (Okolab) at 37 °C, 5% CO$_2$, by an Andor Dragonfly 502 spinning-disc confocal microscope for 7 days.

## High-resolution 4D ovaries live-cell long-term imaging

To observe the detailed behaviors and development of oocytes in ovaries during the perinatal period, images were acquired by an Andor Dragonfly 502 spinning-disc confocal microscope equipped with a 40 × 1.3 NA, a sCMOS camera (Andor Zyla 4.2), and 488-nm (mG) and 568-nm (mT) lines of the Andor ILE system with a spinning-disc confocal scan head (Andor Dragonfly 502) (Zhang et al, 2021). The ovaries were acquired through Z-step mode with the index. In detail, images were acquired with laser 488-nm around 15 to 25%, laser 568-nm around 15 to 20%, exposure time 100 to 200 ms, and Z-step 0.7 μm for 30 to 50 μm by Fusion 2.1 software (https://andor.oxinst.com/products/dragonfly#fusion). To record the developmental dynamics of oocytes, 10-12 fields

(308.72 × 308.72 μm²/field) in four to six ovaries were acquired at 1.5 interval hours for 162 continuous hours (Figs. 2A–F and 6A; Appendix Fig. S2) or at 15 interval min for 12 h staring from c-PD1 (Fig. 3D; Appendix Fig. S4B–E) or for capturing the static images at c-PD1, c-PD2 or c-PD4 (Appendix Fig. S7A–C; Fig. EV4A). Then, the z-stack images were merged and analyzed by Imaris software (https://imaris.oxinst.com) to reconstitute 3D or 4D live oocyte developmental images.

To obtain the subcellular structures on oocytes in the imaging, the Z-stack images of time-lapse imaging were projected for about 30 μm by Imaris or ImageJ software (http://rsbweb.nih.gov/ij) to reconstitute 3D oocyte live images. The ≥2.5 μm filopodia structure on the oocyte was defined as the FL structure. The ODs were identified from time-lapse imaging, which showed an oocyte burst of at least three debris (the diameter of ODs from 1.2 to 14 μm).

## Histological analysis and immunofluorescence staining

For histological analysis, the ovaries were fixed in 4% paraformaldehyde (PFA, Santa Cruz, 30525-89-4) for 8 hr at 4 °C, dehydrated in ethanol and xylene, embedded in paraffin, then sectioned serially at 8 μm with a microtome (RM2245, Leica), deparaffinized and rehydrated. To observe ovarian morphology, sections were stained with hematoxylin (Santa Cruz, sc-24973A).

For immunostaining, the ovarian sections were first treated at high temperature (95–98 °C) for 16 min in 0.01% sodium citrate buffer (pH 6.0) to retrieve antigen. Then, 10% donkey serum (017-000-121, Jackson ImmunoResearch) was used to block the sections for 60 min at room temperature. Ovarian sections were incubated with different primary antibodies overnight at 4 °C. The primary antibodies were as follows: FOXL2 antibody (IMG-3228, goat, 1:300, Novus), DDX4 antibody (ab27591, mouse, 1:200, Abcam) and TOM20 antibody (#42406, Rabbit, 1:200, Cell Signaling Technology). After PBS buffer washed, the sections were incubated with Alexa Fluor 555- or 488-conjugated donkey secondary antibody (1:100, Invitrogen) for 90 min at room temperature. Nuclear was counterstained by Hoechst 33342 (B2261, 1:100, Sigma). Images were acquired on a Nikon Eclipse Ti digital fluorescence microscope or an Andor Dragonfly spinning-disc confocal microscope. To count the oocyte number, tissue sections were stained with DDX4 to label the oocytes, and the number of DDX4-positive oocytes were counted in every fifth section and then multiplied by five to determine the total number of oocytes in each ovary. Five ovaries were counted in each group for the statistical analysis.

## Ovarian allo-transplantation

Ovarian allo-transplantation was performed according to previously described protocols (Zhang et al, 2020b). Generally, the cultured ovaries (cultured from 13.5 dpc to c-PD4) were transplanted under the kidney capsules of bilaterally ovariectomized adult females (6 to 8 weeks). The recipient females were anesthetized with Avertin (300 mg/kg; T48402, Sigma-Aldrich) and the bilateral ovaries were surgically removed. Then, the kidneys were exposed, and the capsule were cut to form an approximately 1-mm wound for seeding the tissues. The cultured ovarian was gently implanted into the kidney capsule through the wound for further in vivo development. After 2 or 4 weeks of surgery, the

transplanted ovaries were collected to detect the development of the ovaries and ovarian follicle distribution.

To detect the developmental potent of oocytes from transplanted ovaries, the recipient females were intraperitoneal injected 5 IU pregnant mare serum gonadotropin (PMSG, Sansheng Biological Technology) followed by 5 IU human chorionic gonadotropin (hCG, Sansheng Biological Technology) to stimulate ovulation after 4 weeks of ovarian transplantation. After 10 h of hCG treatment, the transplanted ovaries were collected and the cumulus oocyte complexes (COCs) were puncture obtained from Graff's follicles, and then in vitro fertilization (IVF) were performed.

## In vitro fertilization assay

The cumulus oocyte complexes (COCs) were collected as described above, and digested with 0.01% hyaluronidase (H-3757, Sigma) to gain cumulus-free oocytes. In vitro fertilization assay was performed as previously described protocols (Zhang et al, 2020a). The cumulus-free oocytes in HTF medium (MR-70-D, Merk-Millipore) were co-incubated with capacitated epididymal sperm of wild-type male mice in an incubator at 37 °C and 5% $CO_2$. Zygotes were washed and transferred to KSOM medium (MR-020P-5F, Merk-Millipore) for culturing to later stages after incubation for 6 h. The embryos at different stages were acquired by Hoffman Modulation Contrast microscope (OLYMPUS IX71).

## High-resolution imaging of the subcellular structure of oocytes in the fresh ovaries

To detect the labeling efficiency of oocytes in the *Oct4-CreER^{T2};mTmG* ovaries after Tam treatment, female gonads were collected at 11.5 dpc (Fig. EV2A,B) or 13.5 dpc (Appendix Fig. S1B; Fig. EV2B) after different dosages of Tam injection at 10.5 dpc. To observe FLs and ODs in vivo, the *Oct4-CreER^{T2};mTmG* ovaries were collected at PD1 (Fig. 3F) after 30 mg·kg^{−1} BW of Tam injection at 12.5 dpc. The gonads and ovaries were fixed in 4% paraformaldehyde for 8 h at 4 °C and washed with PBS containing 0.2% Triton X-100 and 0.5% 1-thioglycerol in the dark for 24 h at room temperature. Next, the ovaries were incubated in the clearing medium (40% *N*-methylacetamide (M26305, Sigma-Aldrich) supplemented with 86% Histodenz (D2158, Sigma-Aldrich), 0.1% Triton X-100 and 0.5% 1-thioglycerol (M1753, Sigma-Aldrich) in the dark at room temperature on a rotor for 72 h as previously described protocols (Li et al, 2017; Xu et al, 2022). The clear *Oct4-CreER^{T2};mTmG* fluorescent gonads and ovaries were acquired by an Andor Dragonfly spinning-disc confocal microscope with a 40× 1.3 NA Z-step 0.8 μm for a 250-μm working distance objective as previously described indexes. Images were acquired by Fusion 2.1 software (https://andor.oxinst.com/products/dragonfly#fusion). For showing the detail morphology of FLs and ODs in vivo, the magnification images (Fig. 3F, middle and right) were cropped from the 3D reconstructed image of the whole transparent ovary.

## Mitochondria detection in cultured ovaries

To label mitochondria in oocytes in live ovaries, the *Oct4-CreER^{T2};mTmG* ovaries at c-17.5 dpc were incubated with MitoTracker™ Orange CMTMRos (1:1000, M7510, Thermo Fisher Scientific) at 37 °C, 5% $CO_2$. After mitochondria staining, the ovaries were washed with DMEM-

F12 medium with 10% FBS and then imaged by an Andor Dragonfly spinning-disc confocal microscope as previously described indexes.

## Transmission electron microscope imaging

The cultured ovaries at c-19.5 and c-PD2 and fresh ovaries at PD1 were fixed in 1% paraformaldehyde with 2.5% glutaraldehyde, and then transferred to 1% osmium tetroxide for 2 h at room temperature. After fixation, the samples were embedded in Spurr's epoxy resin. Ultrathin sections were cut using an ultramicrotome (Leica EM UC7) and stained with both 2% uranyl acetate and lead citrate. The ultrastructure of oocytes was observed under a transmission electron microscope (Hitachi HT7800). Mitochondria in oocytes and OD were captured. Image data were analyzed using ImageJ software (http://rsbweb.nih.gov/ij/).

## Total mitochondrial numbers quantification

The number of mitochondria per germ cell at different ages was counted and calculated using TEM images, as described in a previous report (Lei and Spradling, 2016). Oocyte sections, selected based on the maximal cross-sectional area of the nucleus, were examined, and the number of mitochondria was counted (Nmit_s). The diameters of approximately 100 mitochondria were also measured at each stage to calculate the average mitochondrial diameter (Dave_mit). The volume of oocytes was measured by *Oct4-CreER^{T2};mTmG* oocytes at each stage was calculated based on confocal images. The diameters of 100 oocytes were measured to calculate the average radius of oocytes (Rave_oo), the volume of oocytes (Voo), Voo = 4/3*π*Rave oo^3 and the surface area of the largest cross-section (Sloo), Sloo = 3.14* Rave oo^2. The total number of mitochondria in an oocyte (Nmit_total) was then subsequently calculated by using the equation: Nmit_total = Voo /(Sloo* Dave_mit)* Nmit_s. About 40 oocytes at each stage were measured.

## Single-cell dissociation of ovarian tissue

The approach of single-cell dissociation was modified from a previously published method (Mostovoy et al, 2016). The ovaries from C57 mice at PD1 were collected and cut into small pieces. Then the ovarian pieces (~5 ovaries/tube) were treated with 0.125% Trypsin (25200056, Thermo Fisher Scientific) and 5 IU/ml RNase-free DNase I (EN0523, Thermo Fisher scientific) at 37 °C for total 15 min (5 min × 3), then neutralized in DMEM-F12 medium (11320-033, Gibco) with 10% FBS (16000-044, Gibco). After digestion, the mixture was filtered through a 40-μm cell strainer (352340, BD Biosciences) to avoid any cell aggregations, and then the cells were centrifuged at 300×g for 5 min. Removed the supernatant and resuspended the cell-precipitation with DMEM-F12 medium and 10% FBS. To separate somatic cells and oocytes as described protocol (Teng et al, 2015), the ovarian single-cell mixture in a 35 mm dish was incubated at 37 °C, 5% $CO_2$ for 8 h. The supernatant containing oocytes was harvested and centrifuged at 300×g for 5 min. Removed the supernatant and resuspended the cell-precipitation with DMEM-F12 medium (1% BSA).

## RNA-sequencing and primary sequencing analysis

For single-cell RNA-sequencing, oocytes at PD1 were harvested as described above. Then, cell viability and concentration were measured

by LUNA (LUNA-FL™, Logos Biosystems) after AO/PI Cell Viability Kit staining (F23001, Logos biosystems). The oocytes' viability was 85.9% and the cell concentration was 1330 cells/µl. Then the cells were processed following 10x Genomics protocol (Mostovoy et al, 2016). In brief, single-cell gel bead-in-emulsions (GEMs) were generated by the Chromium Controller instrument (10x Genomics). After GEM-reverse transcriptions (GEM-RTs), GEMs were collected, and the cDNAs were amplified and cleaned up with the SPRIselect Reagent Kit (B23318, Beckman Coulter). The libraries were constructed using the Chromium Single-Cell 3′ Library Kit version 3.1 (10x Genomics, Dual Index). Sequencing libraries were generated by NEBNext® UltraTM RNA Library Prep Kit for Illumina® (NEB, USA) and sequenced on an Illumina Novaseq platform by Beijing Novogene (Beijing, China).

## Single-cell RNA-seq data processing

Barcode processing and gene counting were performed by Cell Ranger (v5.0.1) count pipeline with default parameters. Briefly, reads after trimming were mapped to the mm10 mouse genome (https://10xgenomics.com/) and the genome annotation (gencode vM23) with STAR packaged in the Cell Ranger, and then aligned reads were filtered for identifying cell barcodes and unique molecular identifiers (UMIs). As a result, 6093 barcodes with a median of 9879 UMIs per cell were obtained. The valid cell barcodes and UMIs were used for identifying cell clusters.

## Cluster and DEG analysis in single-cell sequencing analysis

The R package Seurat (v4.1.0)(Satija et al, 2015) were used to identify cell clusters and markers. First, cells were filtered by three criteria: the number of detected genes was greater than 100 and less than 7500. The percentage of mitochondrial genes was less than 15%. For subsetting oocytes, the counts of *Ddx4* and *Dazl* oocyte genes was greater than 0 and the counts of *Wt1* and *Lgr5* somatic cell genes were less than 2. After the filter, 1307 cells were retained. Then, data normalization and scaling were performed using the functions NormalizeData and ScaleData. The UMI counts of each gene were divided by the total UMI counts for that cell, multiplied by 10,000, and after natural-log transformation, the normalized data were scaled and centered. Then, the scaled data were performed dimensionality reduction using the function RunPCA based on principal component analysis (PCA). Using a permutation test method based on null distribution (function ElbowPlot), we selected the top 20 PCA dims to cluster cells by function FindNeighbors and FindClusters based on the shared nearest neighbor (SNN) algorithm. To visualize the cell clusters, the function RunTSNE was performed. Then, for each cluster, genes with an average $\log_2$-transformed difference greater than 0.25, a *P* value less than 0.01, and expressed in at least 25% cells were identified as cluster markers using the function FindAllMarkers based on the Wilcoxon rank sum test. These cluster markers (Dataset EV1) were compared with widely accepted markers to help determine the cell identities of each cluster.

## Pseudotime analysis

The R package monocle (v2.18.0)(Qiu et al, 2017) were used to infer the potential differentiation trajectory of ovary mesenchymal cells. Briefly, after dimensionality reduction based on DDRTree algorithm, high variable genes identified by function differentialGeneTest were used to define cells in a pseudotime trajectory. All cells were divided into three states and ordered in three branches based on reversed graph embedding (RGE). The branch, mostly made up of C0 and C1 was designated as root state, and then branch-related genes (Dataset EV2) among high variable genes were shown in the heatmap using function plot_genes_branched_-heatmap. The expression dynamics through pseudotime of representative genes were shown in Appendix Fig. S9B using function plot_genes_branched_pseudotime.

## GO enrichment analysis

The R package clusterProfiler (v3.18.1) (Yu et al, 2012) was used to perform GO enrichment analysis. Multiple hypothesis test correction method was set to "BH", and the *P* value and *q* value cutoff were set to 0.01 and 0.05, respectively. The results of GO enrichment analysis were attached in Datasets EV3 and 4 and representative GO terms were shown in the corresponding figures.

## Investigations of the mechanisms regulating oocyte phagocytosis

In the autophagy inhibitors or Formin inhibitor treatment experiments, after 4 days of adherent culture, the *Oct4-CreER^{T2};mTmG* or *Oct4-CreER^{T2};Rainbow* ovaries at c-17.5 dpc were treated with or without autophagy signaling cascade inhibitors (5 mM 3-MA, M9281, Sigma-Aldrich; 400 nM MRT68921, S7949, Selleckchem; 0.2 nM Bafilomycin A1, HY-100558, MedChemExpress; 4 µM Chloroquine, HY-17589A, MedChemExpress) or Formin inhibitor (20 µM SMIFH2, HY-16931, MedChemExpress) for 3 or 7 days, which media was changed every other day. The culture-ovaries were imaged by an Andor Dragonfly spinning-disc confocal microscope as previously described indexes.

To check the developmental potent of oocytes, the 3-MA treated ovaries were washed by DMEM-F12 medium at c-PD4, and continued cultured in DMEM/F12 supplemented with 10% FBS and 1% ITS without inhibitor. The oocytes in cultured ovaries were then acquired by Nikon Eclipse Ti digital fluorescence microscope every other day for 7 days for measuring the size changes of oocytes.

## Statistical analysis

Regarding the statistical analysis of the survival ratio of oocytes, as mentioned in the manuscript, a total of 1099 EGFP-positive oocytes in six labeled ovaries were imaged from c-17.5 dpc and traced for 162 h at 1.5-h intervals. Throughout the live imaging, some oocytes moved out of view due to the developing nature of the ovaries and the active movement of the oocytes. The complete developmental dynamics of 594 out of the 1099 oocytes were successfully captured in the six labeled ovaries. The survival ratio of oocytes in each ovary was counted to calculate the final survival ratio of oocytes in the 4D imaging system. (Related to Fig. 2A,B; Appendix Fig. S4A)

To identify the patterns of oocyte death, shrinkage is identified by the oocyte gradually decreasing in size over time, while degradation is characterized by the formation of several small debris. Totally six ovaries were analyzed to quantify the proportion

of oocyte elimination patterns, with more than 15 oocyte eliminations per ovary were traced (Related to Fig. 2C; Appendix Fig. S3B).

To quantify the percent of single cell in live ovaries, we counted the numbers of separated single cell in every 24 h (from c-17.5 dpc to c-PD1) of total oocytes in high (Fig. 2G) or low (Fig. 2I) tamoxifen induced $Oct4\text{-}CreER^{T2};mTmG$ ovaries. In details, the movement of oocytes was continuously traced in reconstituted 4D images every 24 h from c-17.5 to c-PD1 within six ovaries for Fig. 2G ($n = 6$ ovaries, with more than 40 oocytes per ovary; total tracked oocyte numbers: c-17.5 dpc: $n = 320$ oocytes; c-18.5 dpc: $n = 289$ oocytes; c-19.5 dpc: $n = 256$ oocytes; c-PD1: $n = 247$ oocytes) and within six ovaries for Fig. 2I ($n \geq 6$ ovaries, with more than 15 oocytes per ovary; total tracked oocyte numbers: c-17.5 dpc: $n = 321$ oocytes; c-18.5 dpc: $n = 258$ oocytes; c-19.5 dpc: $n = 135$ oocytes; c-PD1: $n = 93$ oocytes). Oocytes with potential connections (as shown in Fig. EV2D) were excluded from the count of single cells. To determine the ratio of single oocytes, each 24-hour interval was treated as a separate imaging unit. The number of single oocytes was divided by the total number of oocytes at the start of this period to calculate the ratio of single oocytes (Related to Fig. 2G,I).

To analyze the correlation of the formation of FL and OD structure, we quantified the difference value of FL ratio (the numbers of oocyte with FLs/total oocytes) and the frequency of oocyte death events (the numbers of oocyte death events/total oocytes) in every 24 h from c-17.5 dpc to c-PD4 (Fig. 3C). In detail, identification of oocytes with FLs and oocyte death was based on continuous tracing of 4D images. Each 24-h interval was treated as a separate period for counting, and the number of oocytes exhibiting FLs or experiencing cell burst was divided by the total oocyte count at the beginning of each period to obtain the corresponding ratio. (Related to Fig. 3C)

To measure the extension dynamics of FL on surviving oocyte, we performed in-depth analysis of high-resolution 4D imaging at the c-PD1 with short intervals (15 min). 20 surviving oocytes that displayed clearly visible FL structures were randomly selected from three ovaries for tracing, ensuring each FL structure's developmental dynamics were traced de novo for accurate and unbiased data collection. The developmental dynamics of newly formed FL structures were meticulously traced over time. The extension speed and the duration of the active extension periods of the FL structures were measured (Related to Appendix Fig. S4B,C).

To analyze the correlation of surviving oocyte growth and phagocytosis events in Fig EV6F, we counted the total number of FLs on the oocyte of 33 image frames during c-19.5 to c-PD2 (as shown in x-axial) and measured the ratio of the diameter of each oocyte at c-PD2/c-19.5 dpc (as shown in y-axial). The fitted curve ($y = -4E\text{-}05x^3 + 0.0034x^2 - 0.0431x + 1.3572$) was generated by Excel. (Related to Appendix Fig. S4F)

To measure the correlation between oocyte survival and the frequency of engulfment events, we tracked the development of 107 surviving oocytes within six ovaries from over a fixed 48-hour period (c-19.5 dpc to c-PD2) and tracked their fate at c-PD4. The engulfment event was identified by the continuous behavior of OD enclose and disappear in the time-lapse imaging. (Related to Appendix Fig. S5)

To quantify the mixture cytoplasm of oocytes, we counted the percent of the numbers of purple oocytes in total oocytes every 24 h from c-17.5 to c-PD4 in six traced $Oct4\text{-}CreER^{T2};Rainbow$ ovaries. Each 24-h interval was treated as a separate period for counting, and the number of purple oocytes was divided by the total oocyte count at the beginning of each period to obtain the corresponding ratio. (Related to Fig. 4C)

To identify the distribution of mitochondria in oocytes and ODs, we measured the area of mitochondria in oocytes (Fig. 4G) and the ratio of the mitochondria area/each oocyte or OD area (Fig. 4H). To determine mitochondrial density in the ODs, we meticulously tracked the entire developmental process of 23 individual sacrificed oocytes across eight areas from four ovaries, resulting in the formation of a total of 107 ODs. Subsequently, we calculated the ratio of mitochondrial area to oocytes or ODs, generating data on relative mitochondrial density (Fig. 4H) (Related to Fig. 4G,H).

In the experiments to investigate the mechanisms regulating oocyte phagocytosis, the numbers of oocytes exhibiting FLs and ODs were counted in the 3D static reconstructed c-PD1 ovaries derived from the long-term 4D time-lapse imaging. The ratio of FLs and ODs were calculated by dividing the number of FLs or ODs by the total number of oocytes. Statistical analysis was performed on the ratios obtained from 4-6 traced $Oct4\text{-}CreER^{T2};mTmG$ ovaries. (Related to Appendix Fig. S7B; Fig. EV4B; Appendix Fig. S15G,H)

To assess the effects of 3-MA for oocyte phagocytosis, we detected a series state indexes of oocytes with or without 3-MA treatment, including the percent of oocyte with FLs/total oocytes from c-18.5 dpc to c-PD4 (Fig. 6C), the diameter of oocytes every 24 h c-17.5 to c-PD4 (Fig. 6D) and the frequency of oocyte death events (the numbers of oocyte death/total oocytes) in every day from c-19.5 dpc to c-PD2 (Fig. 6E). To count the frequency of oocyte death (Fig. 6E), each 24-h interval was treated as a separate period for counting, and the number of oocytes experiencing cell burst was divided by the total oocyte count at the beginning of each period to obtain the corresponding ratio. Totally four ovaries in control and four ovaries in 3-MA treatment group were used in this measurement, and more than 400 oocytes were traced and counted in each time point (Related to Fig. 6C–E).

All experiments were repeated at least three times. Sample organism participants were randomly allocated into experimental groups. In the process of analyzing experimental results, there was no blind assignment of researchers. Data were presented as the mean ± SD or SEM of each result. Two group data were calculated by Student's $t$-test, and were regarded statistically significant at $P < 0.05$. $P$ is suggested as follows: $*P < 0.05$, $**P < 0.01$, $***P < 0.001$ and not significant (n.s.), $P \geq 0.05$. More than two group data were calculated by ANOVA tests. $P$ (a, b) < 0.05, $P$ (a, c) < 0.05, $P$ (b, c) < 0.05. Statistics and charts were gained by using Prism 5 (GraphPad Software, La Jolla).

# Data availability

The raw sequence data reported in this paper have been deposited in the Genome Sequence Archive in National Genomics Data Center (GSA: CRA012740) that are publicly accessible at https://ngdc.cncb.ac.cn/gsa/browse/CRA012740. The source data of this paper are collected in the following database record: biostudies:S-SCDT-10_1038-S44318-025-00430-3.

The source data of this paper are collected in the following database record: biostudies:S-SCDT-10_1038-S44319-025-00663-7.

## Code availability

The raw sequence data reported in this paper have been deposited in the Genome Sequence Archive in National Genomics Data Center (GSA: CRA012740) that are publicly accessible at https://ngdc.cncb.ac.cn/gsa/browse/CRA012740. The source data of this paper are collected in the following database record: biostudies:S-SCDT-10_1038-S44318-025-00430-3.

## Peer review information

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

## Acknowledgements

The authors are grateful to Yun Li, Shan Jiang, Dr. Lan Jiang (Beijing Institute of Genomics, Chinese Academy of Sciences) for scRNA-seq technology support, Dr. Kui Liu (Department of Obstetrics and Gynecology, The University of Hong Kong, Hong Kong, China) for kindly sharing the *Rosa26^{rbw/+}* mice and Dr. Ping Zheng (Yunnan Key Laboratory of Animal Reproduction, Kunming Institute of Zoology, Chinese Academy of Sciences, Kunming, China) for kindly sharing the *Oct4-CreER^{T2}* mice. This study was supported by the National Key Research and Development Program of China to HZ and YZ (2022YFC2703800 and 2024YFC2706600), the Key Program of National Natural Science Foundation of China to HZ (82230051), the National Natural Science Foundation of China to YZ (82371664), the 2115 Talent Development Program of China Agricultural University to YZ (1021-00109022), The Innovative Project of State Key Laboratory of Animal Biotech Breeding to HZ (2023SKLAB1-6), and the Space Application System of China Manned Space Program to HZ (KJZ-YY-NSM0609).

## Author contributions

**Yan Zhang**: Conceptualization; Resources; Data curation; Formal analysis; Supervision; Funding acquisition; Validation; Investigation; Visualization; Methodology; Writing—original draft; Project administration; Writing—review and editing. **Yingnan Bo**: Investigation; Visualization. **Kaixin Cheng**: Investigation; Visualization. **Ge Wang**: Formal analysis; Investigation. **Lu Mu**: Data curation. **Jing Liang**: Data curation. **Lingyu Li**: Investigation. **Kaiying Geng**: Investigation. **Xuebing Yang**: Investigation. **Xindi Hu**: Investigation. **Wenji Wang**: Formal analysis. **Longzhong Jia**: Investigation. **Xueqiang Xu**: Formal analysis; Investigation. **Jingmei Hu**: Writing—review and editing. **Chao Wang**: Writing—review and editing. **Fengchao Wang**: Resources; Investigation. **Yuwen Ke**: Formal analysis. **Guoliang Xia**: Project administration. **Hua Zhang**: Conceptualization; Resources; Data curation; Supervision; Funding acquisition; Validation; Investigation; Methodology; Writing—original draft; Project administration; Writing—review and editing.

Source data underlying figure panels in this paper may have individual authorship assigned. Where available, figure panel/source data authorship is

listed in the following database record: biostudies:S-SCDT-10_1038-S44319-025-00663-7.

## Disclosure and competing interests statement

The authors declare no competing interests.

# Expanded View Figures

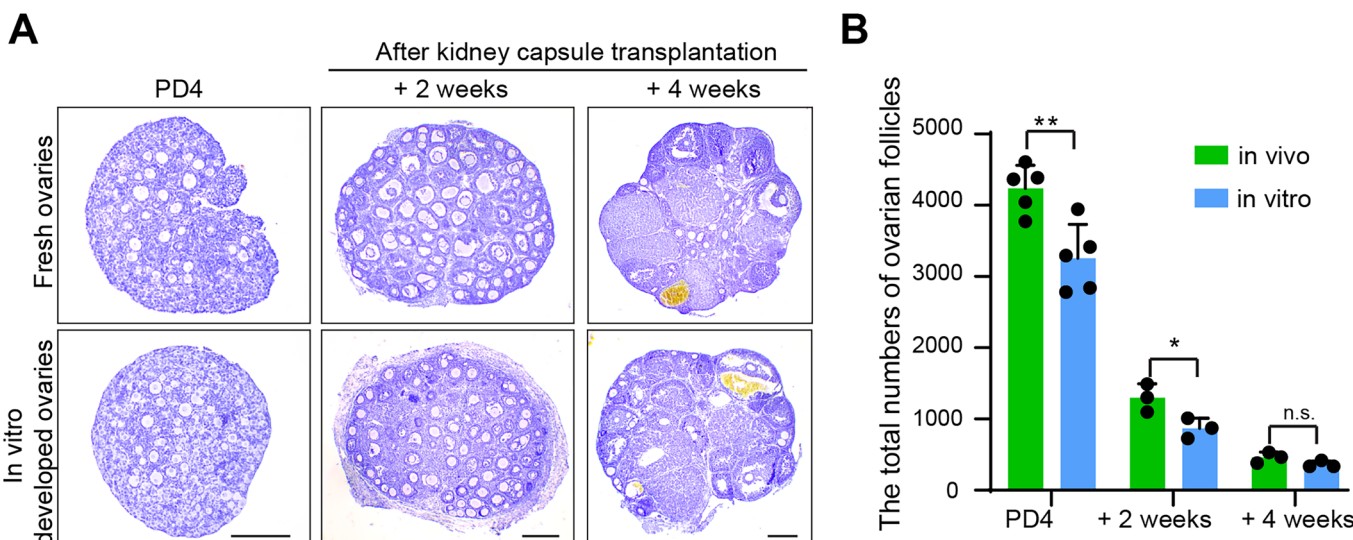

**Figure EV1. Normal follicle development in cultured ovaries after allo-transplantation.**

(A) Histological analysis displaying a regular distribution of follicles in both fresh and cultured ovaries throughout different developmental stages after kidney capsule transplantation. Scale bar: 100 μm. (B) Follicle counting detection revealed a slight reduction in the number of follicles in cultured ovaries compared to those in fresh ovaries at c-PD4 and 2 weeks post-transplantation. $n \geq 3$ ovaries at every time point. The data were presented as the mean ± SD. Statistical significance is determined using a two-tailed unpaired Student's $t$-test; PD4: $p$ value = 0.0065; 2 weeks: $p$ value = 0.013; 4 weeks: $p$ value = 0.148. n.s. $P > 0.05$, *$P \leq 0.05$, **$P \leq 0.01$.

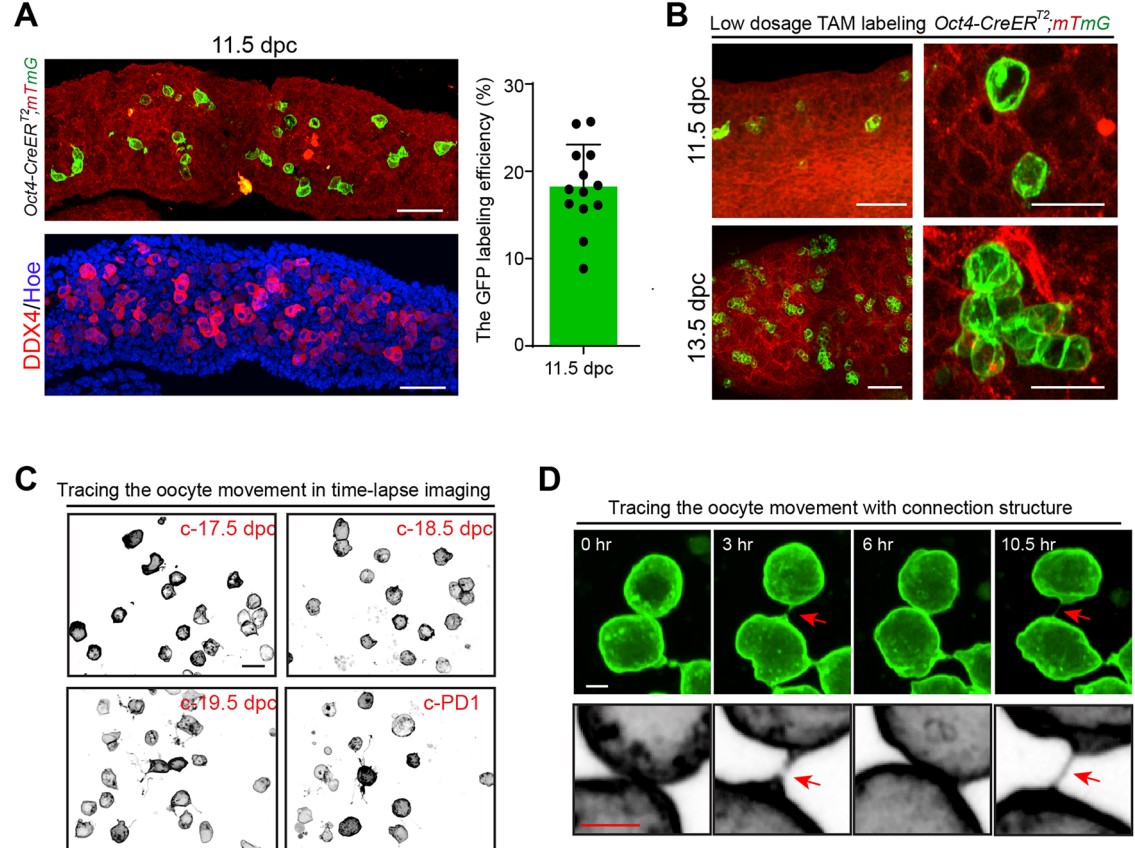

**Figure EV2. Tracing the oocyte movement and separation in the *Oct4-CreER^T2;mTmG* ovaries after low dosage Tam treatment.**

(A) Evaluation of the labeling efficiency in oocytes within *Oct4-CreER^T2;mTmG* ovaries at 11.5 dpc. Pregnant females carrying *Oct4-CreER^T2;mTmG* fetus were treated with tamoxifen (20 mg·kg⁻¹ BW) at 10.5 dpc (left upper). Immunofluorescent staining of oocytes using DDX4 antibody in the same sections (left bottom). Statistical analysis of oocyte labeling efficiency, determined by the ratio of GFP-positive oocytes to the total number of DDX4-positive cells. Data collected from 13 sections across four ovaries are presented as mean ± SD (right). (B) Showing the labeling efficiency of oocytes in the *Oct4-CreER^T2;mTmG* ovaries. Showing the labeled single germ cells at 11.5 dpc and separated cysts at 13.5 dpc in the ovaries after a low dosage of Tam treatment. Scale bar: 50 μm. (C) Tracing the movement of labeled oocytes demonstrated majority of oocytes moving as single cells from c-17.5 dpc to c-PD1. Oocytes were displayed in inverted black/white (b/w) to highlight. Scale bar: 20 μm. (D) Showing the criteria for identifying single oocytes. Oocytes with any potential connections (arrows) were excluded from the counting of single cells. Scale bar: 5 μm.

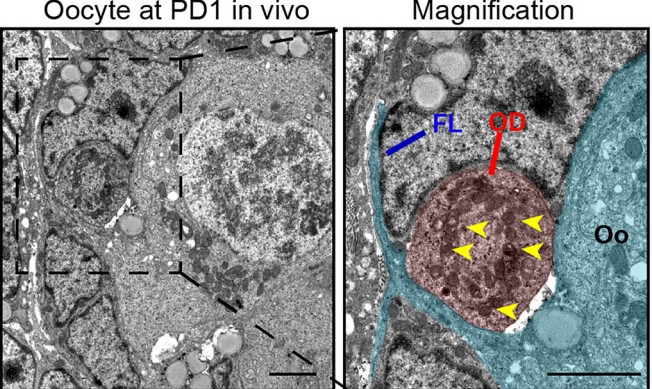

**Figure EV3. Oocyte phagocytosis is observed in a TEM image at PD1.**

Representative TEM image of an oocyte at PD1 in the ovary, showing an oocyte (cyan) containing FL structures (blue) and mitochondria (yellow arrowheads) within an OD (red). Scale bar: 2 μm.

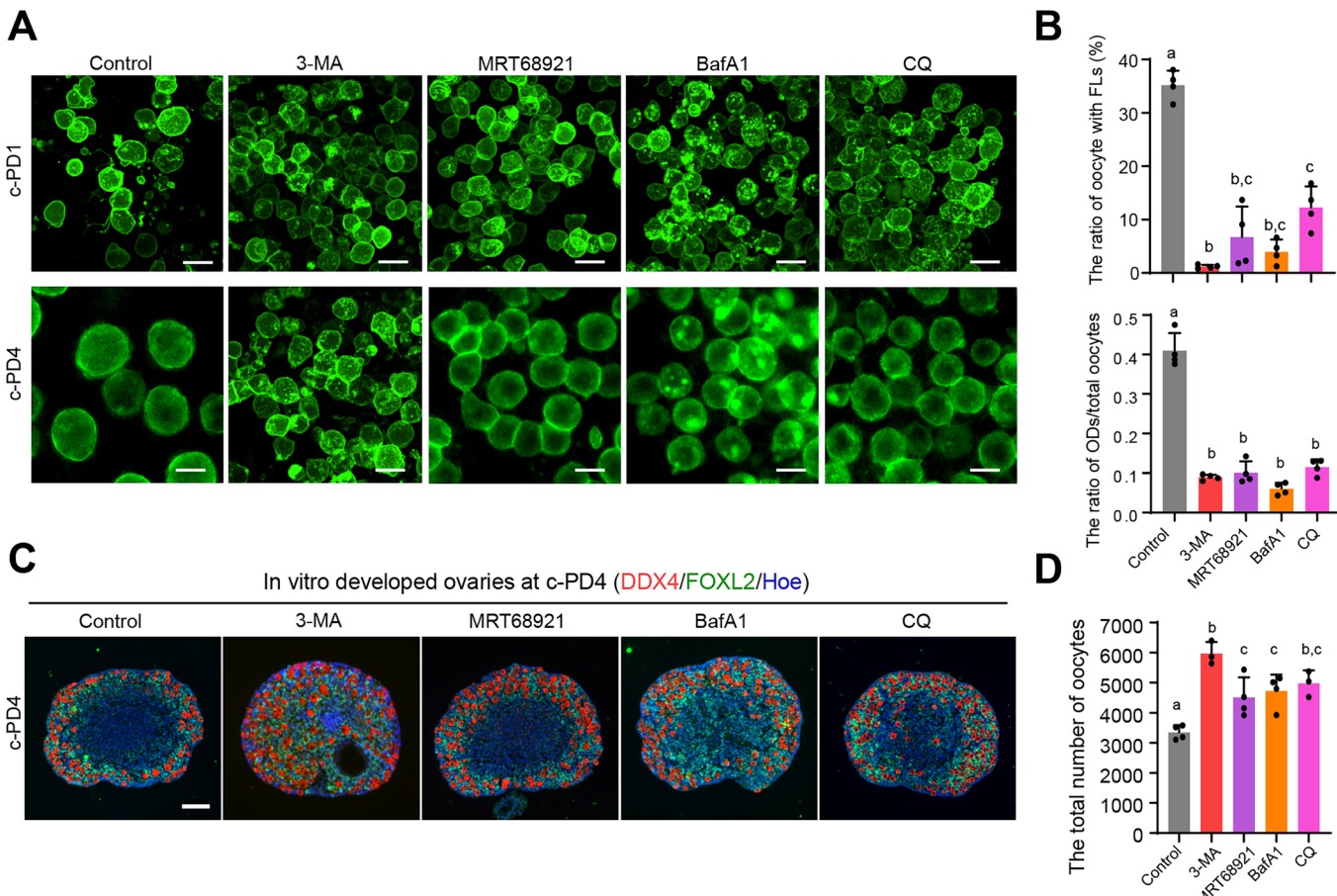

**Figure EV4. Impact of autophagy inhibition on oocyte phagocytosis during ovariogenesis.**

(**A**) The analysis of oocyte phagocytosis and development in cultured ovaries treated from c-17.5 to c-PD4 with autophagy inhibitors (3MA, MRT68921, BafA1, and CQ). Autophagy inhibition consistently reduced sacrifice and phagocytosis at c-PD1 and increased the survival of oocytes by c-PD4. Scale bar: 20 μm. (**B**) Statistical evaluations showed that suppression of autophagy significantly diminishes oocyte sacrifice and phagocytosis at c-PD1, demonstrating a sharp decline in the ratio of surviving oocytes with FLs and a decrease in the formation of ODs from sacrificed oocytes. The analysis included four ovaries per group, evaluating more than 70 oocytes per ovary. FL: control vs. 3-MA: $p$ value = 5.9E-09; control vs. MRT: $p$ value = 6.7E-08; control vs. BafA1: $p$ value = 1.9E-08; control vs. CQ: $p$ value = 1.1E-06; 3-MA vs. MRT: $p$ value = 0.21; 3-MA vs. BafA1: $p$ value = 0.79; 3-MA vs. CQ: $p$ value = 0.003; MRT vs. BafA1: $p$ value = 0.80; MRT vs. CQ: $p$ value = 0.22; BafA1 vs. CQ: $p$ value = 0.05. OD: control vs. 3-MA: $p$ value = 2.3E-10; control vs. MRT: $p$ value = 4.1E-10; control vs. BafA1: $p$ value = 7.2E-09; control vs. CQ: $p$ value = 7.8E-08; 3-MA vs. MRT: $p$ value = 0.96; 3-MA vs. BafA1: $p$ value = 0.54; 3-MA vs. CQ: $p$ value = 0.66; MRT vs. BafA1: $p$ value = 0.23; MRT vs. CQ: $p$ value = 0.94; BafA1 vs. CQ: $p$ value = 0.07. (**C, D**) Immunofluorescent staining of the ovarian morphology (**C**) and oocyte number counting (**D**) at c-PD4 showed an increased number of surviving oocytes in ovaries treated with autophagy inhibitors. Red - DDX4; Green - FOXL2; Blue - Hoechst. Scale bar: 100 μm. Data were from at least three ovaries per group. Data were presented as the mean ± SD. control vs. 3-MA: $p$ value = 7.25E-05; control vs. MRT: $p$ value = 0.034; control vs. BafA1: $p$ value = 0.011; control vs. CQ: $p$ value = 0.0054; 3-MA vs. MRT: $p$ value = 0.012; 3-MA vs. BafA1: $p$ value = 0.035; 3-MA vs. CQ: $p$ value = 0.155; MRT vs. BafA1: $p$ value = 0.96; MRT vs. CQ: $p$ value = 0.71; BafA1 vs. CQ: $p$ value = 0.95. Statistical significance was determined by ANOVA tests. $P$ (a, b) < 0.05, $P$ (a, c) < 0.05, $P$ (b, c) < 0.05.

