## [Peer Review File · EMBO Reports]

Cyst-independent oocyte phagocytosis builds the female reproductive reserve in mice

Hua Zhang, Yan Zhang, Yingnan Bo, Kaixin Cheng, Ge Wang, Lu Mu, Jing Liang, Lingyu Li, Kaiying Geng, Xuebing Yang, Xindi Hu, Wenji Wang, Longzhong Jia, Xueqiang Xu, Jingmei hu, Chao Wang, Fengchao Wang, Yuwen Ke, and Guo-liang Xia

Corresponding author(s): Hua Zhang (huazhang@cau.edu.cn) , Yan Zhang (yanzhang1011@cau.edu.cn)

Review Timeline:	Transfer Date:	11th Aug 25
	Editorial Decision:	5th Sep 25
	Revision Received:	15th Sep 25
	Accepted:	11th Nov 25

Transaction Report:

The first two review rounds of this manuscript were performed in another journal. A revised version of this manuscript was transferred to EMBO reports following peer review at the EMBO Journal.

Referee #1:

During female germ cell development in the fly, incomplete cytokinesis following mitosis generates 16-cell cysts connected by cytoplasmic bridges. Fifteen of the cells, designated nurse cells, transfer their cytoplasm to the single 'winner' cell that becomes the oocyte. Spradling's group has recently provided evidence that a similar system operates in mammals. Here, the authors propose an additional (or alternative) possibility - that certain oocytes phagocytose fragments of other, dying, oocytes thereby increasing their size, and that this phagocytosis is essential to produce healthy oocytes. This is a dramatic and exciting claim that if correct would re-write textbooks. As such, however, it requires robust supporting evidence. Unfortunately, although the authors have done a tremendous amount of work, the evidence presented is not nearly sufficient to support the claim. In addition, the authors have not fully responded to other substantive issues raised in previous reviews.

Three main lines of evidence are presented to support the claim.

First, in an ingenious approach, the authors use the Rainbow transgenic line to show that two fluorochromes are present in the cytoplasm of some oocytes after in vitro culture of the ovaries. This is most easily explained by phagocytosis. Three issues, however, arise. First, the number of two-color oocytes does not exceed about 20% (Fig. 4C, S7D). This raises the question of how widespread the phenomenon is. Second, and more importantly for their model, this experiment does not test whether phagocytosis is necessary for oocytes to survive. Third, also crucially, the authors state in their response to one reviewer that, in vivo, 'we found that all oocytes expressed only a single type of fluorescent protein and no purple oocytes were observed in the ovaries at PD5 (Fig. R1-3)'. Taken at face value, this argues against cytoplasmic mixing via phagocytosis in vivo. It is possible that phagocytosis occurred several days earlier and the phagocytosed fluorescent protein was subsequently degraded. But it is unavoidable that this result is a major concern for the model. One way to resolve this would be to inject Tamoxifen into the Rainbow mice at a time after cysts have broken down and fix the ovaries at different time-points (eg, d-19, PD-1), then stain with antibodies against the GFP, RFP, CFP. If they see double-labeling of many oocytes, this will be strong evidence for cytoplasmic sharing via phagocytosis independently of cysts.

Second, the authors show that the number of mitochondria increases during the period of time when phagocytosis occurs. There is no evidence, however, to reject the simple interpretation that oocytes are manufacturing new mitochondria. In other words, this observation provides no evidence to support the authors' model.

Third, the authors show that phagocytosis and oocyte growth are blocked when ovaries are treated with drugs that block autophagy. Their interpretation is that blocking autophagy prevents phagocytosis which prevents oocyte growth. This rests on the assumption that blocking autophagy has no harmful effect on the cells other than to prevent phagocytosis. However, autophagy is a normal event in cells and blocking it, particularly for several days as has been done here, may seriously damage the cells. It is worth noting also that the main drug used, 1-MA, blocks type III PI-3 kinase and so will inhibit any reaction requiring this enzyme. As designed, the experiment cannot discriminate between these multiple potential explanations for the observations.

In sum, while the model is interesting and may be correct, the results unfortunately do not provide rigorous experimental support for it.

Additional points and concerns:

The authors state that the oocytes became assembled into primordial follicles, as shown in Fig. 1G. In Fig. 2A, however, we cannot see any red granulosa cells around the green oocytes, though red cells are visible in the background. This raises the concern that the oocytes being imaged are those that have not been assembled into primordial follicles. Please explain why no granulosa cells are detectable. More generally, it is important to indicate what fraction of the oocytes became assembled into follicles in their culture system. Was it a common event or a rare event?

A previous reviewer emphasized the importance of distinguishing between cysts, where cells are connected by cytoplasmic bridges, and nests, where unconnected cells are clustered together. The imaging techniques here do not allow one to determine whether neighboring oocytes are part of the same cyst or are in a nest. Despite this point raised by the reviewer, the authors continue to use the word cyst throughout the manuscript. See l. 182-186 for an example.

Throughout the manuscript, the authors write that some oocytes are sacrificed through a process of selection to improve the quality of the remaining oocytes. See l. 21, 23, 59, 104 for examples. I am not aware of any experimental evidence supporting this claim. The authors need either to provide specific citations or to rephrase.

l. 26 - It has not been shown here that phagocytosis is required for oocyte survival.

l. 106 - Authors write that phagocytosis is required to select the best oocytes. But surviving oocytes phagocytose fragments of dead oocytes. So selection, if it occurs, precedes

phagocytosis.

Fig. 3D, E - How do we know that the engulfed fragment is part of an oocyte? Why is it not enclosed by a fluorescent membrane?

l. 246-251 - Authors have not addressed a previous criticism that surviving oocytes may engulf more fragments because they live longer than dead oocytes, not because engulfing a minimum number of fragments is required for survival.

Minor points

Please remove all subtitles and descriptive from figures, as per standard formatting practice. Use a uniform font and point size for axis labels.

Fig. 2G, 2I, 3C, 4C, 6C-E. X-axis should indicate number of days in culture, not the inferred equivalent in vivo.

Although the response to reviews and new text (in red) in the manuscript are written in excellent English, few edits have been done elsewhere in the text, as shown by the absence of red text that would show changes. Much work remains, including in the Figures, to improve the quality of the English.-

Referee #2:

I saw significant improvement in the manuscript, and this manuscript can be a strong candidate for the EMBO journal in its current form.

Dear Hua,

Thank you for the transfer of your manuscript from The EMBO Journal to EMBO reports. Your manuscript has been peer reviewed at another journal, which could not offer publication. You have subsequently submitted your further revised version to The EMBO Journal and the handling editor contacted former referee #2 (now #1) and former referee #4 (now #2). While referee #2 considered the revisions adequate and supported publication, referee #1 had a number of remaining concerns.

I have discussed your study and the remaining concerns with my colleague Ieva Gailite at The EMBO Journal and we have subsequently invited you to transfer your study to EMBO reports with a view towards rapid publication with textual changes.

I had gone through the remaining concerns of Referee #1 once more and note here again what I considered to be the most important ones in a short summary:

1) Data from Rainbow mouse shows oocytes with double-fluorescence - phagocytosis is plausible but:

- Double-labelling only seen for 20% of oocytes, so how widespread is this phenomenon?
- not tested whether phagocytosis is necessary for oocytes to survive.
- Double-labelling in vivo not observed. This argues against cytoplasmic mixing via phagocytosis in vivo.

2) Number of mitochondria increases

- possibly through phagocytosis but oocytes might simply manufacture new mitochondria.

3) Phagocytosis and oocyte growth are blocked when ovaries are treated with drugs that block autophagy

- could simply be caused by a general detrimental effect of autophagy inhibition. 1-MA also blocks type III PI-3 kinase, could have autophagy-independent effects.

I think that all these concerns can be addressed in the text by a thorough discussion of the evidence at hand, its limitations and the possibility of alternative scenarios. All conclusions need to be carefully phrased, with your data supporting your hypothesis but not ruling out to a 100% other, confounding factors or alternative explanations.

You have meanwhile provided a point-by-point response to these concerns, i.e., to my summary above. As discussed, I invite you to revise your study for publication in EMBO reports. Please address these points in your manuscript text and please submit it with track changes or marked up text, to speed up the editorial evaluation.

Please carefully phrase your conclusions and discuss alternative scenarios in the manuscript text. I understand that you tested the role of phagocytosis using inhibitors of autophagy and lysosomal function, but do agree that strictly speaking, these experiments do not rule out that a more general perturbation of autophagy in surviving oocytes causes the observed phenotypes. While inhibition phagocytosis is a valuable and maybe most likely scenario to explain the effect on oocyte survival, a general effect of bulk or selective autophagy inhibition cannot be ruled out with a 100% certainty, in my opinion, and this possibility should be discussed.

Your manuscript will be sent to an expert advisor for a read-through and final approval of the textual changes.

=====

In addition to these textual changes addressing remaining referee concerns, there are also a number of things we need from the editorial side:

- Please remove the keywords from the manuscript text and upload these as a separate file.
- We noted a discrepancy in author names: it is Wenji Wang in the manuscript text vs JiWen Wang in the online manuscript tracking system.
- Regarding the Author Contributions, we now use CRediT to specify the contributions of each author in the journal submission system. Therefore, please remove the Author Contributions from the manuscript file and make sure that the author contributions in our online manuscript tracking system are correct and up-to-date. The information you specified in the system will be automatically retrieved and typeset into the article. You can enter additional information in the free text box provided, if you wish.
- Datasets: Dataset EV1 still has Table S1 in its file name (see tab when you open the file); consider adding the legends directly to the excel files in a separate tab/worksheet. In either case, please remove the red font.

- EV movies: the red font in the legends can be removed
- Appendix: please remove the red font in the figure legends.
- Appendix Figure S8, S10: please specify the p-value for *** in panels A-C and the statistical test used.
- Appendix: please specify the exact p-values if $p > 0.05$ or $p < 0.01$. This is not required for very small p-values of < 0.001 or < 0.0001).
- Please upload the Reagents and Tools Table as a separate file and remove it from the manuscript text. It will be added back by our typesetters.
- Data Availability section: Please provide URLs that resolve directly to the dataset(s) at GSA.
- Figures should be called out in a numerically linear manner. In this context we note that Fig 6F is called out before Fig 3F.
- We perform a routine data integrity check on all figures and .xls files. Doing so, we noticed the following points that need clarification:
 1. The oocytes shown in Figure 2E have been reused in Figure 2F (0 hr, 13.5 hr). Please clearly state this re-use in the figure legend.
 2. The timelapse images shown in Figure 2B are also part of the series shown in Appendix Fig S2. Please clearly state this reuse in the respective figure legends.
 3. The same is true for Figure 2C and Appendix Fig S3B. But here the timing seems not to match. For "Shrinkage" the 34.5 hr image in 2C is the + 0 hr timepoint in S2B and 37.5 hrs in 2C is +3 hr in S2B. Similar for "Degradation" (58.5 hr is +0 hr, 63 hr is +1.5 hr). Please explain the reuse and the different timing in the figure legends.
 4. The images shown in Figure 2B at 70.5 and 79.5 hrs are used again in Appendix Fig S5A, but here with the time stamps 72 hr and 81 hr. Moreover, the cells seem to have been horizontally flipped to match. The other images from this time series appear different. Please check.
 5. There appears to be a cell reuse between Appendix Figure S2 and Appendix Figure S5A:
 - S2 "70.5 hr" is similar to Appendix Fig S5A, "72 hr"
 - S2 "85.5 hr" is similar to S5A "67. 5 hr"
 - S2 "79.4 hr is similar to S5A 81 hr
 Again, the images seem to have been horizontally flipped to match.

Please provide an explanation for the reuse and flipping.

- Our production/data editors have asked you to clarify several points in the figure legends (see below). Please incorporate these changes in the manuscript and return the revised file with tracked changes with your final manuscript submission.
 1. Please indicate the statistical test used for data analysis in the legend of figure 5F.
 2. Please note that information related to n is missing in the legend of figure 5D.
 3. Please note that the white arrows are not defined in the legend of figures 2C, D. This needs to be rectified.
 4. Please note that the blue arrows are not defined in the legend of figure 2F. This needs to be rectified.

- Finally, please upload the synopsis image as a separate file.

With kind regards,

Martina

Point-by-Point response

1) Data from Rainbow mouse shows oocytes with double-fluorescence - phagocytosis is plausible but:

- Double-labelling only seen for 20% of oocytes, so how widespread is this phenomenon?

Author response: Thank you for the suggestion. Our previous report showed that the recombination efficiency of the Rainbow model using Tamoxifen treatment is relatively low, which makes it difficult to label all oocytes in Rb mice (Zhang et al., PNAS, 2012). Our strategy was to label primordial germ cells (PGCs) that contain only one Rainbow allele, leading to cysts that include 8-32 oocytes, all expressing the same fluorescent protein. Thus, when oocyte phagocytosis occurs, most neighboring oocytes derived from the same cyst will share the same fluorescence, and only those at the boundary of two cysts will display different colors, resulting in the low mixing ratio we observed. Furthermore, the purple fluorescence from dual-recombination at DNA would be permanent, while the cytoplasmic exchange-derived purple fluorescence is only a temporary mixture, which subsequently degraded. Our Rainbow experiments demonstrate that cytoplasm exchange does not rely on connections and attachments between oocytes after the cyst breakdown, suggesting that oocyte phagocytosis is the key mechanism for cytoplasm and organelle enrichment during the final period of oocyte selection.

In fact, we conducted a systematic quantitative analysis of the frequency of phagocytosis in our study. To evaluate the phagocytosis frequency, we analyzed the engulfment behaviors of 107 oocytes of *Oct4-CreER^{T2};mTmG* across six ovaries within a fixed 48-hour window (c-19.5 dpc to c-PD2) and tracked their survival at c-PD4 (Fig. R1-1) in the round 1 revision of *Nature* rebuttal. Crucially, we excluded engulfment events occurring after c-PD2 to isolate the effect of early engulfment on subsequent survival. We found that all surviving oocytes had phagocytosed oocyte debris (ODs). Our analysis revealed a strong correlation between the number of engulfed OD fragments and oocyte survival: all oocytes that engulfed ≤ 4 OD fragments during this period failed to survive at c-PD4, whereas those that formed follicles had significantly higher engulfment activity (averaging 15 OD fragments). This suggests that a threshold level of engulfment is necessary, though not solely sufficient, for oocyte survival.

For these reasons, we did not include additional discussion on this point in the revised manuscript, as we believe such a discussion would be redundant and detract from the significance of our findings.

Fig. R1-1. Analyzing the relationship between the frequency of engulfment events and the

developmental fate of winner oocytes.

(A) Identification of the frequency of potential engulfment events through an in-depth analysis of the 4D time-lapse images. Both FL connected ODs and the attached ODs were counted as one engulfment event. A surviving oocyte (upper) and an eliminated oocyte (bottom) were shown to illustrate the criteria for identifying engulfment events during the same 48-hour observation window (from c-19.5 dpc to c-PD2). Scale bar: 10 μm . (B) The relationship between the frequency of engulfment events and the fate of oocytes during the 48-hour developmental period. All oocytes that engulfed ≤ 4 OD fragments uniformly failed to survive at c-PD4. In contrast, surviving oocytes that formed follicles had significantly higher engulfment activity, with an average of 15 OD fragments engulfed during this period. $n = 107$ oocytes from 6 ovaries.

- not tested whether phagocytosis is necessary for oocytes to survive

Author response: Thank you for the suggestion. While the Rainbow experiment was not designed to address this specific question, we have indeed found that phagocytosis, as a mechanism of oocyte selection, is essential to boost oocyte quality.

To determine the physiological significance of oocyte phagocytosis in the ovaries, we blocked autophagy using a series of well-tested autophagy inhibitors including 3-methyladenine (3-MA), MRT68921, Bafilomycin A1 (BafA1), and Chloroquine (CQ) from c-17.5 dpc, and traced the oocyte development and fate until c-PD4. In general, we found that the oocytes in all inhibitor treated groups remained in a relatively silenced state with no active oocyte phagocytosis at c-PD1 (Fig 6A-B and Fig EV4A-B). Consequently, more oocytes survived at c-PD4 (Fig 6A-B and Fig EV4C-D), and reflecting a failure of the oocytes to undergo normal programs of either growth (Fig 6D and Fig EV4) or elimination (Fig 6E and Fig EV4). Consistent with the inactivity of oocyte phagocytosis, we found significantly increased number of oocytes at the end of imaging of c-PD4 in 3-MA, MRT68921, BafA1 and CQ treated ovaries (Fig 6F and Fig EV4D).

Next, we extended the time of in vitro culture to monitor the developmental capability of oocytes that lacked the phagocytosis. We traced the growing of oocytes after 7 days washing 3-MA at c-PD4, showing the lacking phagocytosis oocytes grew only to $37.4 \pm 6.1 \mu\text{m}$ (Fig 6G and Appendix Fig S10C, 3-MA) compared to control ($52.4 \pm 4.8 \mu\text{m}$, Fig 6G and Appendix Fig S10C). These data demonstrated that the oocytes that did not undergo phagocytosis related cytoplasm exchange were unable to grow fully. Meanwhile, we performed allo-transplantation to examine the in vivo developmental capability of the oocytes with or without phagocytosis (Fig 6H). Strikingly, no healthy oocytes survived in the treated ovaries after 4 weeks of surgery (Fig 6H, 3-MA + 4 weeks), which sharp contrast to the healthy development and ovulation of oocytes (Fig 6H, control) in the control group. Therefore, we conclude that the oocyte phagocytosis is essential to boost oocyte quality for the maintenance of normal fertility in female mice.

- Double-labelling in vivo not observed. This argues against cytoplasmic mixing via phagocytosis in vivo.

Author response: Thank you for the suggestion. In our experiments, we used heterozygous Rainbow embryos, in which the diploid cells contain only one *Rainbow* allele, to test cytoplasmic exchange. We injected Tam at 10.5 dpc, before the germ cells entered meiosis, to induce

recombination. Technically, this strategy should guarantee that each oocyte expresses only one type of fluorescent protein.

Given that meiosis begins at 13.5 dpc in ovaries, another possibility is that the purple oocytes are derived from the dual-recombination of tetraploid oocytes, which contain two *Rainbow* alleles in the heterozygous *Rainbow* embryos. However, the purple fluorescence from dual-recombination should be permanent, while the cytoplasmic exchange-derived purple fluorescence is only a temporary mixture. To test this, we treated *Oct4-CreER^{T2};Rainbow* females with Tam at 10.5 dpc ($30 \text{ mg}\cdot\text{kg}^{-1}$ BW) and examined the color of oocytes at PD1 (Fig. R1-2, left), when oocyte phagocytosis occurred and PD5 in vivo (Fig. R1-2, right), when cytoplasmic exchange had concluded. We found that about $17.3 \pm 3.6\%$ purple oocytes were observed in ovaries at PD1 (Fig. R1-2, left). However, all oocytes expressed only a single type of fluorescent protein and no purple oocytes ($0 \pm 0\%$) were observed in the ovaries at PD5 (Fig. R1-2, right). This result indicated that phagocytosis occurred several days earlier (19.5dpc-PD2) and the phagocytosed fluorescent protein was subsequently degraded. These data confirm that the formation of purple oocytes is solely due to cytoplasmic exchange.

Fig. R1-2. Fluorescent expressions in the oocytes of *Oct4-CreER^{T2};Rainbow* ovaries at PD1 (left) and PD5 (right), showing the formation of purple oocytes is solely due to cytoplasmic exchange. Scale bar: 100 μm .

2) Number of mitochondria increases

- possibly through phagocytosis but oocytes might simply manufacture new mitochondria.

Author response: Thanks for the valuable suggestions. We have expanded our discussion of this point “While mitochondrial fission may also contribute to the increase in oocyte mitochondrial population during this stage of selection, it is clear that oocyte phagocytosis is one of the main mechanisms driving the enhancement of selected oocyte quality.” in the revised manuscript line 449-452.

3) Phagocytosis and oocyte growth are blocked when ovaries are treated with drugs that block autophagy

- could simply be caused by a general detrimental effect of autophagy inhibition. 1-MA also blocks type III PI-3 kinase, could have autophagy-independent effects.

Author response: Thanks for the valuable suggestions. We also have expanded our discussion to explain the oocyte disappear after inhibiting autophagy: ‘However, our findings also reveal the complexity of autophagy's involvement in oocyte phagocytosis. Notably, the upregulation of five ATG8 family proteins, including ATG3, is not restricted to core autophagy pathways but also extends to alternative mechanisms such as LC3-associated phagocytosis and responses to lysosomal damage (Jacomin et al, 2020). This suggests that these alternative autophagy-related pathways, rather than canonical autophagy, may be more directly involved in regulating the sacrificed oocytes. Furthermore, the use of broad-spectrum autophagy inhibitors complicates our interpretation, as they affect both surviving and sacrificed oocytes, thereby masking the specific role of autophagy in shaping the loser phenotype.’ in the discussion (line 485-494).

Point-by-point responses to the manuscript format changes:

- Please remove the keywords from the manuscript text and upload these as a separate file.

Author response: Thank you for the suggestion. We have removed the keywords and upload the revised manuscript as a sperate file. The revised words have labeled with yellow highlight.

- We noted a discrepancy in author names: it is Wenji Wang in the manuscript text vs JiWen Wang in the online manuscript tracking system.

Author response: Thank you for the suggestion. We have modified the name of Wenji Wang in the system.

- Regarding the Author Contributions, we now use CRediT to specify the contributions of each author in the journal submission system. Therefore, please remove the Author Contributions from the manuscript file and make sure that the author contributions in our online manuscript tracking system are correct and up-to-date. The information you specified in the system will be automatically retrieved and typeset into the article. You can enter additional information in the free text box provided, if you wish.

Author response: Thank you for the suggestion. We have removed the author contributions from the manuscript and added these information in the CRediT of the system.

- Datasets: Dataset EV1 still has Table S1 in its file name (see tab when you open the file); consider adding the legends directly to the excel files in a separate tab/worksheet. In either case, please remove the red font.

Author response: Thank you for the suggestion. We have merged the legend to Excel file in a separate sheet and revised the file name and color font.

- EV movies: the red font in the legends can be removed

Author response: Thank you for the suggestion. We have revised the red font to black.

- Appendix: please remove the red font in the figure legends.

Author response: Thank you for the suggestion. We have revised the red font to black in the figure legends.

- Appendix Figure S8, S10: please specify the p-value for *** in panels A-C and the statistical test used.

Author response: Thank you for the suggestion. We have added the p-value and the statistical test in FigS8 and S10.

- Appendix: please specify the exact p-values if $p > 0.05$ or $p < 0.01$. This is not required for very small p-values of < 0.001 or < 0.0001).

Author response: Thank you for the suggestion. We have added the exact p-value in Fig S11 legend.

- Please upload the Reagents and Tools Table as a separate file and remove it from the manuscript text. It will be added back by our typesetters.

Author response: Thank you for the suggestion. We have removed the Reagents and Tools Table

from manuscript and uploaded this as a separated file.

- Data Availability section: Please provide URLs that resolve directly to the dataset(s) at GSA.

Author response: Thank you for the suggestion. We have added the URLs of our single cell RNA-sequences dataset in the manuscript.

- Figures should be called out in a numerically linear manner. In this context we note that Fig 6F is called out before Fig 3F.

Author response: Thank you for the suggestion. We apologize for the error in figure citation. We have corrected the figure citation to 'Appendix Figure S4F' in Line 229.

- We perform a routine data integrity check on all figures and .xls files. Doing so, we noticed the following points that need clarification:

1. The oocytes shown in Figure 2E have been reused in Figure 2F (0 hr, 13.5 hr). Please clearly state this re-use in the figure legend.

Author response: Thank you for the suggestion. The figure 2E is the model illustration to reflect individual oocytes separated, even though seen cyst-like structures with different numbers of connected oocytes in 3D static images of Figure 2F (0 hr and 13.5 hr). The detailed information was in Line 166-172 of manuscript. To avoid misunderstanding, we have added the state "the 4D tracing images (from the time point1 to point2 of Fig 2E)" in the figure 2F legend in Line 1170.

2. The timelapse images shown in Figure 2B are also part of the series shown in Appendix Fig S2. Please clearly state this reuse in the respective figure legends.

Author response: Thank you for your suggestion. Figure 2B shows representative images of surviving oocyte development, which are part of the time-lapse series presented in Appendix Fig S2. To avoid any misunderstanding, we have added the statement, "All images of surviving oocyte development traced in Figure 2B are shown in Appendix Figure S2," to the Figure 2 legend (lines 1163–1164).

3. The same is true for Figure 2C and Appendix Fig S3B. But here the timing seems not to match. For "Shrinkage" the 34.5 hr image in 2C is the + 0 hr timepoint in S2B and 37.5 hrs in 2C is +3 hr in S2B. Similar for "Degradation" (58.5 hr is +0 hr, 63 hr is +1.5 hr). Please explain the reuse and the different timing in the figure legends.

Author response: Thank you for your suggestion. The data in Appendix Fig S3B represent quantification of oocyte elimination events corresponding to lane 2 and lane 3 in Figure 2C. The labels "+0 hr" and "+3 hr" refer to the time points at which oocyte elimination events occurred. To avoid confusion, we have revised these labels to "before" and "after" in Appendix Fig S3B. We have also updated the figure legend to clarify that the quantification in Appendix Fig S3B legend corresponds to the oocyte elimination events shown in Figure 2C (lane 2 and lane 3).

4. The images shown in Figure 2B at 70.5 and 79.5 hrs are used again in Appendix Fig S5A, but here with the time stamps 72 hr and 81 hr. Moreover, the cells seem to have been horizontally flipped to match. The other images from this time series appear different. Please check.

Author response: Thank you for the suggestion. We apologize for the errors in the time labels. The data presented in Appendix Fig S5A represent quantification of the phagocytosis frequency of surviving or sacrificed oocytes, as shown in Figure 2B. Additionally, we had previously flipped the images for clarity in labeling the OD fonts. To avoid any confusion, we have now corrected the time labels and restored the original orientation of the images in Appendix Fig S5A. We have added the statement in Appendix Fig S5A legend.

5. There appears to be a cell reuse between Appendix Figure S2 and Appendix Figure S5A:

S2 "70.5 hr" is similar to Appendix Fig S5A, "72 hr"

S2 "85.5 hr" is similar to S5A "67.5 hr"

S2 "79.4 hr" is similar to S5A 81 hr

Again, the images seem to have been horizontally flipped to match.

Please provide an explanation for the reuse and flipping.

Author response: Thank you for your suggestion. We apologize for the errors in the time labels. The data presented in Appendix Fig S5A represent quantification of the phagocytosis frequency of surviving or sacrificed oocytes, as shown in Figure 2B. Additionally, we had previously flipped the images for clarity in labeling the OD fonts. To avoid any confusion, we have now corrected the time labels and restored the original orientation of the images in Appendix Fig S5A. We also have added the statement in Appendix Fig S5A legend. Furthermore, regarding the concern that "Figure S2 85.5 hr is the same as S5A 67.5 hr," this was due to an error in figure placement. We have reviewed and corrected this mistake in Appendix Figure S2.

- Our production/data editors have asked you to clarify several points in the figure legends (see below). Please incorporate these changes in the manuscript and return the revised file with tracked changes with your final manuscript submission.

1. Please indicate the statistical test used for data analysis in the legend of figure 5F.

Author response: Thank you for the suggestion. We have added the statistical test method in the legend of fig 5F.

2. Please note that information related to n is missing in the legend of figure 5D.

Author response: Thank you for the suggestion. We have added the information of n in the legend of Fig 5D.

3. Please note that the white arrows are not defined in the legend of figures 2C, D. This needs to be rectified.

Author response: Thank you for the suggestion. We have added the white arrows defined in the Fig 2C.

4. Please note that the blue arrows are not defined in the legend of figure 2F. This needs to be rectified.

Author response: Thank you for the suggestion. We have added the cyan arrows defined in the Fig 2F.

- Finally, please upload the synopsis image as a separate file.

Author response: Thank you for the suggestion. We have upload the file of synopsis image to the submission system.

Hua Zhang
China Agricultural University
College of Biological Sciences
China

Dear Hua,

We have now received feedback from the expert advisor we asked to assess and read through the final version of your manuscript. Given the positive comment, I am very pleased to accept your manuscript for publication in the next available issue of EMBO reports. Thank you for your contribution to our journal.

Kind regards,

Martina

Referee #1:

I was asked for an additional opinion on this manuscript after peer-reviewing. In my opinion, the authors have obtained an impressive amount of data in a technical tour de force. They show convincingly through live imaging that dominant oocytes absorb cell debris from oocytes that underwent autophagy. I think the point-by-point reply to the comments of one reviewer is satisfying. The only experiment I would suggest is to stain for markers of autophagy and phagocytosis, which should be rather straightforward and a direct proof of phagocytosis. But overall, I think this manuscript should be published in Embo Reports. It raises a lot of questions that -without any doubt- will be the subject of future studies.
